# 4DVAR assimilation of GNSS zenith path delays and precipitable water into a numerical weather prediction model WRF

Witold Rohm[1], Jakub Guzikowski[2], Karina Wilgan[1,4], Maciej Kryza[3]

[1]Institute of Geodesy and Geoinformatics, Wroclaw University of Environmental and Life Sciences, Wroclaw, 50-357, Poland
[2]Institute of Geophysics, Polish Academy of Sciences, Ksiecia Janusza 64, 01452 Warsaw, Poland
[3]Institute of Geography and Regional Development, University of Wroclaw, Wroclaw, 50-137, Poland
[4]Institute of Geodesy and Photogrammetry, ETH Zürich, Zürich, 8093, Switzerland

*Correspondence to*: Witold Rohm (witold.rohm@upwr.edu.pl)

**Abstract.** The GNSS data assimilation is currently widely discussed in the literature with respect to the various applications in meteorology and numerical weather models. Data assimilation combines atmospheric measurements with knowledge of atmospheric behavior as codified in computer models. With this approach, the 'best' estimate of current conditions consistent with both information sources is produced. Some approaches allow assimilating also the non-prognostic variables, including remote sensing data from radar or GNSS (Global Navigation Satellite System). These techniques are named variational data assimilation schemes and are based on a minimization of the cost function, which contains the differences between the model state (background) and the observations. The variational assimilation is a first choice for data assimilation in the weather forecast centres, however current research is consequently looking into use of iterative, filtering approach such as Extended Kalman Filter (EKF).

This paper shows the results of assimilation of GNSS data into numerical weather prediction (NWP) model WRF (Weather Research and Forecasting). The WRF model offers two different variational approaches: 3DVAR and 4DVAR, both available through WRF Data Assimilation (WRFDA) package. The WRFDA assimilation procedure was modified to correct for bias and observation errors. We assimilated the Zenith Total Delay (ZTD), Precipitable Water (PW), radiosonde (RS) and surface synoptic observations (SYNOP) using 4DVAR assimilation scheme. Three experiments have been performed: 1) assimilation of PW and ZTD for May and June of 2013, 2) assimilation of: PW alone; PW, with RS and SYNOP; ZTD alone; and finally ZTD, with RS and SYNOP for 5-23 May, 2013, and 3) assimilation of PW or ZTD during severe weather events in June 2013. Once the initial conditions were established, the forecast was run for 24 hours.

The major conclusion of this study is that for all analyzed cases, there are two parameters significantly changed once GNSS data are assimilated in the WRF model using GPSPW operator and these are: moisture field and rain. The GNSS observations improves forecast in the first 24 hours, with strongest impact starting from 9h lead time. The relative humidity forecast in a vertical profile after assimilation of ZTD shows over 20% decrease of mean error starting from 2.5 km upward. Assimilation of PW alone does not bring such a spectacular improvement. However, combination of PW, SYNOP and radiosonde improves distribution of humidity in the vertical profile by maximum of 12%. In analyzed three severe weather

cases PW always improved rain forecast and ZTD was always reducing humidity field bias. Binary rain analysis shows that GNSS parameters have significant impact of rain forecast in the class above 1mm/h.

## 1 Introduction

The data assimilation in weather forecasts is one of the key component in all prediction systems as it is an initial value problem and the quality of the initial field has large impact on the forecasts. Currently, the leading weather agencies assimilate operationally few dozens of observation data types such as: radiosonde (RS) profiles, radiances from satellite observations, SYNOPs, refractivities from radio occultations, pilot reports and many others (Barker et al., 2004). With an advent of European Cooperation in Science & Technology (COST) actions 716 (1999-2004), 1206 (2013-2017), as well as the project funded in the 5th framework program Targeting Optimal Use of GPS Humidity Measurements in Meteorology" (TOUGH), the adoption of the ground based Global Navigation Satellite Systems (GNSS) observations to the operational forecasts by most of the weather services in Europe become a fact. In this study the term GNSS covers all navigation systems used world-wide, whereas the term Global Positioning System (GPS) is related to only one source of observations – the US based GPS. There are many publications related to either 1) performance of large scale weather forecast systems augmented with many observations including GNSS, 2) added value of GNSS observations in nowcasting services, or 3) case-based studies showing impact of GNSS data in particular cases. The following three approaches are discussed below.

A very comprehensive study done by (Poli et al., 2007) on the global forecast model Arpage using 4 dimensional variational assimilation (4DVAR) shows that the impact of GPS Zenith Troposphere Delay (ZTD) on forecasts is different in winter (improving pressure), spring (reducing surface humidity Root Mean Square Error) and summer (positive impact on wind, geopotential and precipitation, negative on humidity). A similar, very detailed study was done by (Bennitt and Jupp, 2012), where Authors discussed the operational assimilation of GPS ZTD in MetOffice into North Atlantic and European 12 and 24 km model in spring, summer and autumn. The results were mixed: for all cases the introduction of GPS ZTD increased the humidity bias, however the improvements of clouds forecasts were observed. (Bennitt and Jupp, 2012) also identify no clear benefit of 4DVAR against 3DVAR. Lindskog et al., (2017) in their Nordic country study of GNSS ZTD impact on forecasts, confirmed that the forecasts are sensitive to thinning distance. Shorter distance between stations (below 100km) leads to a larger humidity bias in the lower troposphere, which may explain the humidity bias in the (Bennitt and Jupp, 2012) solution. (Lindskog et al., 2017) showed that the humidity forecast is better when the GNSS ZTD is assimilated with other meteorological observations such as Advanced Microwave Sounding Unit (AMSU) or Infrared Atmospheric Sounding Interferometer (IASI) radiances. Authors also showed that the adopted bias correction strategy and GNSS ZTD estimation procedure have marginal impact on the forecasts. All studies run in large weather forecasting systems suggests that the assimilation of GNSS ZTD, either 3D or 4DVAR, on average has mostly neutral impact on the forecast if the system is already saturated with meteorological observations.

Another branch of weather models are those used in nowcasting, such as the (legacy) Rapid Update Cycle RUC 20 km and RUC 40km (Benjamin et al., 2002) or currently operational Rapid Refresh (RAP) (Benjamin et al., 2016) that are targeting short 12h and 24h predictions for decision making and safety operations with a large number of observations assimilated every hour. One of the first experiments using GPS Precipitable Water (PW) in nowcasting service in USA (RUC model
60km) (Smith et al, 2000), showed 1% improvement of the relative humidity forecast in the bottom part of the atmosphere. However, in specific cases related to active frontal weather, the improvement was much larger: 14% in moistening and 24% in drying stage of the advection. The increased spatial resolution to 20km of RUC20 (Smith et al., 2007) shows stronger improvements in humidity field and Convective Available Potential Energy CAPE than with RUC40. The 850 hPa relative humidity (RH) forecasts improve more in the night time, and in the colder season than that in the warmer season. Current
RAP model, running on 13km grid, continuously assimilates GPS PW every hour from 300 stations across US (Benjamin et al., 2016). It shows that there is clear benefit in using GPS observations, especially for short term (nowcasting) predictions.

The third type of studies that are appearing in the literature are case-based, showing the impact of GNSS on particular weather event. One of the first to test the impact of GPS based ZTD observations in Europe was Cucurull et al., (2004).
Authors used NCAR / Penn State Mesoscale Model 5 (MM5) model ZTD 3DVAR assimilation for a case of snow storm in 14-15 December 2001 over western Mediterranean Sea. They found that there are reductions of RMSE wind by 1.7%, temperature by 4.1% and surface humidity by 17.8%. Authors also noted that the forecasts work better if the ground based automatic weather stations were used in the same assimilation run. Another example of an early stage case-based research is the assimilation of GPS PW by Nakamura et al., (2004) with 4DVAR scheme into mesoscale JMA model for summer
intensive rain cases. The assimilation of GPS data improved the precipitation location, but the statistics did not show large improvement. One of the first GPS 4DVAR ZTD study in US was by De Pondeca and Zou, (2001), who run assimilation of GPS observations in MM5 together with the wind profiler data and radio acoustic sounding system (RASS) virtual temperature. Five 12h experiments for California's December frontal system passage were performed. It was found that the ZTD assimilation corrects the underestimation of accumulated rain by 33.15% and 25.08% for 6h and 12h respectively.
Adding the wind profiler improves the forecast by 88.26% and 32.53% and adding further RASS observations increases the performance to 93.21% and 50.58%, respectively. In a more recent study by Boniface et al., (2009), the GPS ZTD was assimilated (3DVAR) for 280 stations over 15 days into high-resolution (2.5km) AROME model. The results were positive for poorly predicted precipitation and neutral for well predicted one. More recently, Tilev-Tanriover and Kahraman, (2014) studied the impact of the GPS PW assimilation in the Weather Research and Forecasting (WRF) model in a 2 days case of
intense snowfall forecasts in the central Anatolia. Authors performed 3 experiments: base run, cold start and cycling all with PW 3DVAR operator. Results show that the cycling assimilation mode decreases the temperature and humidity biases, whereas the cold start performs worse than the control run. Saito et al., (2016) studied the impact of ensemble prediction that did not produce enough precipitation. They found that even downscaling from 10km to 2km, still do not improve locations of precipitation's cores. Finally, the 4DVAR PW assimilation into a non-hydrostatic model improved the location of

scattered intensive rain. In summary, most of the literature reported substantial increase in the quality of the forecast of humidity, rain location and sometimes also the rain accumulated total amount. Less significant improvement was achieved for wind speed and temperature. All studies that used additional observations, especially these resolving vertical structure of the water vapor and temperature, complemented GNSS observations and improved the forecast even more.

The literature review shows that the impact of the GNSS ZTD/PW assimilation depends on the number of already assimilated observations and applied preprocessing (Bennitt and Jupp, 2012; Lindskog et al., 2017; Poli et al., 2007) as well as on the type of weather conditions. The main aim of this paper is to quantify the impact of the GNSS data, both ZTD and PW, gathered operationally in Poland, in weather forecasting. The study is based on the WRF model with high spatial resolution of 4 km x 4 km supported with the WRF Data Assimilation (WRFDA) package. We show the importance of the

GNSS data assimilation for cases of various meteorological conditions observed in May and June 2013, which is a benchmark period for COST Action ES1206. To our best knowledge, no GNSS ZTD/PW assimilation experiment was carried out in Poland yet. Moreover, we found only one publication (Tilev-Tanriover and Kahraman, 2014) dealing with the assimilation of GPS ZTD and PW into widely adopted WRF model using the WRFDA package. We present a study showing the impact of GNSS ZTD and PW observations on the forecasts for a longer time period - two months (May 2013 – calm

weather conditions and June 2013 – active stormy weather), followed by quantifying improvements of adding RS and SYNOP data into the assimilation system already run with GNSS observations and finally we verified impact of GNSS observations on prediction for specific cases.

The paper has following structure: after introduction section short overview of used data and methodology is presented in sections 2 and 3, respectively. These sections are followed by experiments setup description and results (section 4). The

paper is closed with a conclusion section 5.

## 2 Data

The GNSS PW/ZTD data are assimilated into NWP WRF model. The chosen period is covering May and June 2013, with special focus on May, 5-23, 2013 and three shorter cases: a) May, 29-31, 2013, b) June, 17-19, 2013 and c) June, 24-26, 2013. The period is chosen in accordance to the COST Action ES1206 GNSS meteorology benchmark (Douša et al., 2016).

**2.1 WRF model**

In this study the WRF model is used, it is numerical weather prediction system designated for simulation of multiscale, spatial and temporal atmosphere flows. The WRF configuration (Kryza et al., 2013) is based on two nested model domains. The first domain covers the European area with 12km x 12 km grid spacing. The second, nested domain covers Poland and Central Europe with 4km x 4km grid spacing (Fig. 1).

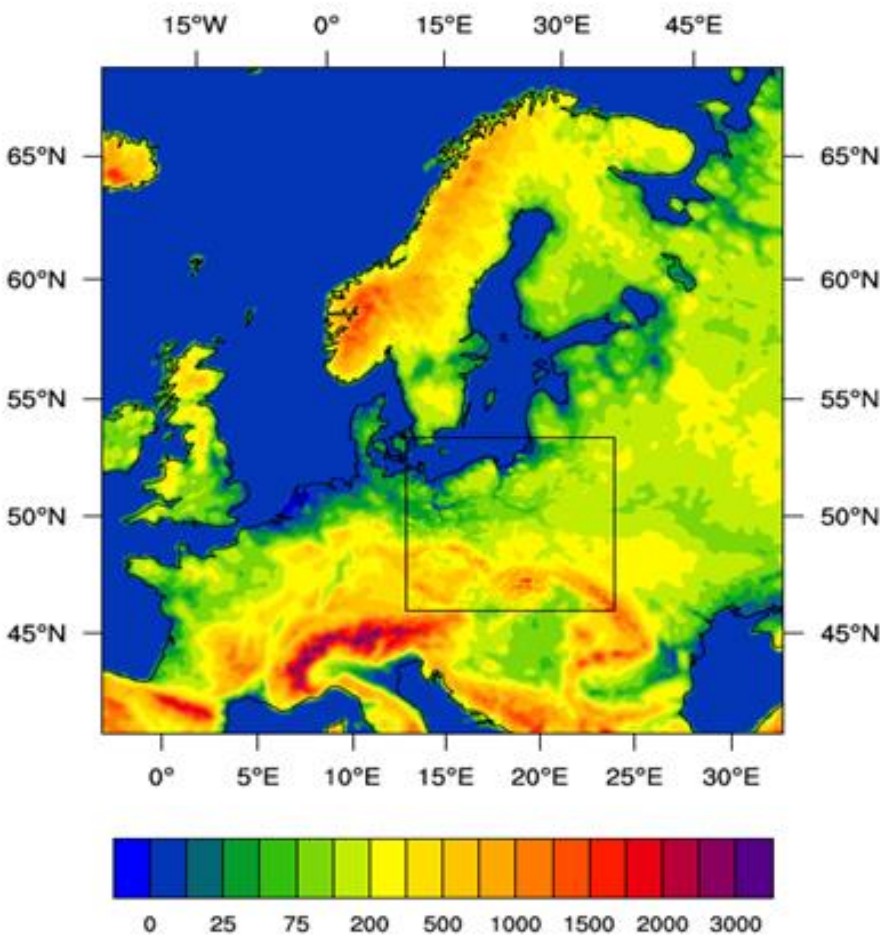

Fig. 1 The WRF model configuration (the inner square represents the nested domain with 4 km x 4 km resolution). Colors denote the orography of the terrain (m a.s.l.).

5    Initial and boundary conditions are taken from the National Center for Environmental Prediction Final Analysis, Operational Model Global Tropospheric Analyses (NCEP FNL) database (National Centers for Environmental Prediction, 2000). The data are available with 1° x 1° horizontal and 6h temporal resolution and with 26 vertical levels from 1000 to 10 hPa. The WRF model for Poland is calculated and provided by the Department of Climatology and Atmosphere Protection of the University of Wroclaw. The details of the WRF configuration are presented in Table 1. Data assimilation was run using
10   4DVAR WRF DA system, only for inner domain (d02). Prediction model was started once a day, at 00 UTC. Assimilation window was centred around 00 UTC. Background Error covariance (BE) was selected for   regional application

(cv_options=5) ( BE depends on the WRF domain). BE was constructed based on a forecast for convection event in the first week of May 2013.

Quality control was selected for SYNOP and RS data in observation processor (obsproc) and in WRFDA. For ZTD and PW data, quality control was conducted before processing, followed by obsproc verification and last step in 4DVAR assimilation. The WRF configuration based on the best ensemble members (ens1) with small modification from ensemble system dedicated for Poland area during summertime (Guzikowski et al., 2017).

## 2.2 GNSS data

The GNSS data are calculated by the GNSS and Meteo working group from Institute of Geodesy and Geoinformatics, Wroclaw University of Environmental and Life Sciences (www.igig.up.wroc.pl/igg). The PW and ZTD values are calculated at 106 stations of the European Position Determination System Active Geodetic Network (ASG-EUPOS, www.asgeupos.pl) in Poland and adjacent areas (Fig. 2).

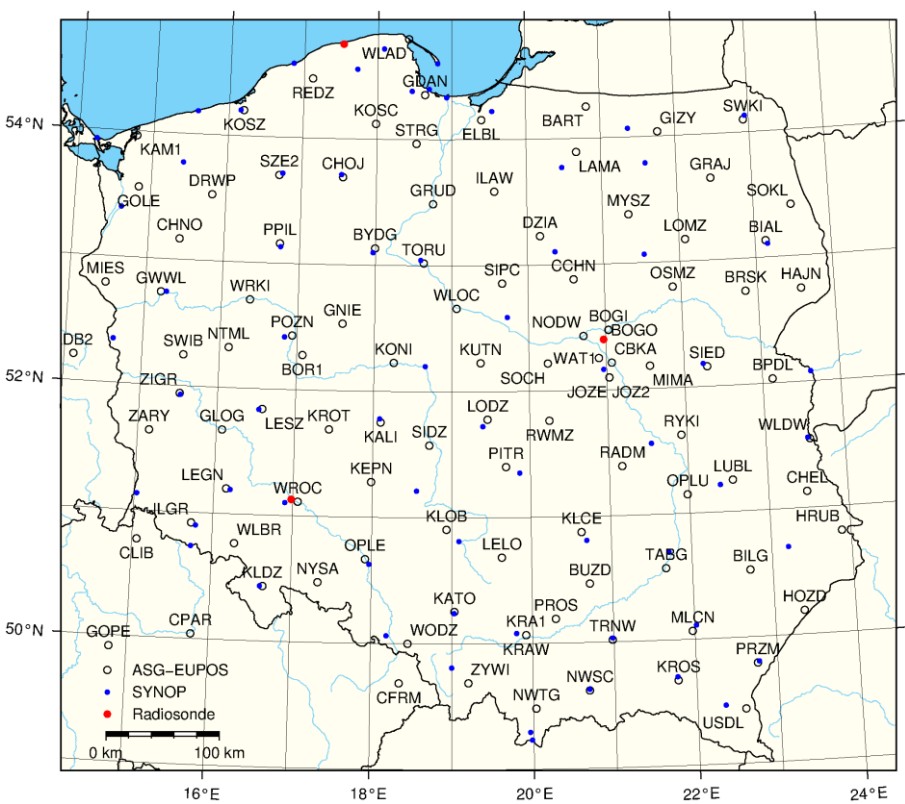

Fig. 2. Location of the GNSS, SYNOP and Radiosonde stations in Poland

The GNSS parameters are calculated from GPS data only, using the Bernese GNSS Software version 5.0 (Dach et al., 2007). The parameters (coordinates and troposphere) are estimated in a near-real time (NRT) regime, 30 min after each full hour, without the gradients estimation. The dry troposphere a-priori model is taken from Saastamoinen, (1972) mapped with Dry Niell MF (Niell, 2000) and the ZTD relative constraining of 3 mm is applied. International GNSS Service (IGS) ultra-rapid orbits, clocks and Earth rotation parameters are used. These parameters are now altered to fit more recent version of Bernese (5.2) (Dach et al., 2015; Dymarska et al., 2017), but this study uses NRT data, originally processed in 2013. This way our impact study will show exactly the minimum potential of GNSS data assimilation in weather model. More details on the GNSS data processing and quality monitoring of the data can be found in Bosy et al. (2012). Fifteen of the stations are a part of the EUREF Permanent Network (EPN) and provide the tropospheric parameters with the accuracy required by NWP data assimilation (Dymarska et al., 2017), i.e. the standard deviation between GNSS ZTD and WRF ZTD is 10 mm and the standard deviation between radiosonde ZTD and WRF ZTD is 14 mm. In the inter-comparison study using multiple techniques (Wilgan et al., 2015), the discrepancy between GNSS observations and radiosonde was found to be 10 mm. According to the EGVAP requirements (Met Office, 2012), this accuracy of the GNSS data is sufficient for the assimilation in NWP models.

## 2.3 Model evaluation

The WRF model runs are compared with: surface meteorological measurements of air temperature, relative humidity, wind speed and precipitation, radiosonde temperature and relative humidity profiles in 3 locations across Poland, GNSS observations in 106 locations.

For the SYNOP measurements are available every hour at 66 SYNOP stations, evenly distributed over the area of Poland, operated by the Institute of Meteorology and Water Management – National Research Institute. Model evaluation is performed only for the nested domain. Four error metrics are calculated to assess the forecast performance:

- Mean error (ME), which describes the model tendency of overestimation (ME >0) or underestimation (ME<0) of the given meteorological parameter. The ME (bias) is calculated as a mean difference between the modelled and observed values for all stations (domain wide). The units are the same as for the analyzed meteorological parameters.

- Root mean squared error (RMSE), which takes only non-negative values. The RMSE (scatter) is calculated as a root of the squared differences between the modelled and observed values for all stations. The units are the same as for the analyzed meteorological parameters.

- Pearson correlation coefficient (corr), which takes values from -1 to +1, and the expected value is 1. Corr is unitless.

- Index of agreement (IOA), developed by (Willmott, 1981) as a standardized measure of the degree of model prediction error. IOA varies between 0 and 1, and 1 indicates a perfect match.

Model performance verification is done using observations for rainfall, wind speed, relative humidity and air temperature at 2 m. The model evaluation is done for each simulation considering entire period and for different lead times and for selected days during which severe weather was observed (case studies). For rainfall forecasts, binary evaluation is also presented using performance diagrams (Roebber, 2009), separately for five different precipitation intensity thresholds. The closer dataset is to the upper right corner of the plot the better performance of the forecasts.

Additionally, the model performance for air temperature and relative humidity was compared with radiosonde data, available with high vertical resolution for three stations located in Poland: Łeba, Warszawa and Wrocław, for the May and June case (large data set) we provide (Fig. 4) profiles mean error with standard deviation multiplied by 1.96 (p=0.05), for other cases (much less observations) only mean errors are provided (Fig. 6, 10, 12). Similar comparison was done by Guerova et al. (2005). The model based PW and ZTD are also compared to the GNSS based retrievals. Bias and standard deviation of the residuals WRF based PW and ZTD minus GNSS observed PW and ZTD are calculated for 106 stations.

## 3 Methodology

The variational assimilation is based on the Bayesian probability theory and it states that the model analysis is inferred from two probabilities: background and observations. These can also be expressed as a minimization of a cost function, with two major components: background B and observations R error covariance (Ide et al., 1997; Lorenc, 1986; De Pondeca and Zou, 2001) in the 4DVAR implementation:

$$J[x(t_o)] = \frac{1}{2}[x(t_o) - x^b(t_o)]^T B_0^{-1}[x(t_o) - x^b(t_o)] + \frac{1}{2}\sum_{i=0}^{n}(H_i[x(t_i)] - y_i^o)R_i^{-1}(H_i[x(t_i)] - y_i^o), \qquad (1)$$

where $x(t_i), x^b, x(t_o)$, are model state vector at the time $t_i$, background vector and model initial conditions $t_0$, respectively. In general case, there are $N$ kinds of observations $y$ defined at discrete times $t_i$ from $t_o$ to $t_n$, where the assimilation window spans from the lowest to the highest $t_i$. The $H_i[x(t_i)]$ is a forward operator that transforms parameters from the model space to the observations space. The 3DVAR differs to 4DVAR by taking $t_i$ equal to observation time and analysis time. Minimization of the equation (1) requires also finding adjoint (ADJ) and tangent linear (TLM) operators, each related to the observation type and forward operator $H_i[x(t_i)]$. For more details, the readers are referred to e.g. Barker et al., (2004) or Huang et al., (2009).

### 3.1 GPSPW operator

The WRFDA system employed in this study hosts variational (VAR) 3D/4D as well as hybrid variational-ensemble algorithms (Barker et al., 2012). Currently, the system supports the assimilation of: surface, radiosonde, aircraft, wind profile observations as well as atmospheric motion vectors, radar reflectivities, spectrometric, GPS radio occultation and GPS ground-based data. The latter is linked directly to the GPSPW operator (The National Center for Atmospheric Research and WRF Model Users' Page, 2017). The operator defines the forward, tangent linear and adjoint of $H$ for the 4DVAR and

3DVAR case for both ZTD and PW. The operator also defines the observation covariance $R$; in here diagonal matrix is assumed, with no correlation between observations, which requires spatial and temporal thinning (Bennitt et al., 2017; Bennitt and Jupp, 2012). The ZTD forward operator $H$ reads as follows (Vedel and Huang, 2004) with further corrections made by Y.-R. Guo (from da_transform_xtoztd module of GPSPW) :

$$ZTD(i,j) = ZHD(i,j) + \sum_{k=kts}^{k=kte} \left( \frac{wdk_1 p(i,j,k)q(i,j,k)}{t(i,j,k)} + \frac{wdk_3 p(i,j,k)q(i,j,k)}{t^2(i,j,k)} \right) \frac{\Delta h}{a_{ew}}, \tag{2}$$

where $i, j, k$ are indices of model nodes, $p$ is a pressure, $q$ is specific humidity, $t$ is temperature, $\Delta h$ is a height difference between two consecutive model layers, $a_{ew} = 0.622$ is a constant, $wdk_1 = 2.21\ 10^{-7}, wdk_3 = 3.73\ 10^{-3}$ are compressibility constants, $ZHD$ is a Zenith Hydrostatic Delay computed according to the (Saastamoinen, 1972) explicitly given in Eq. 6.

The PW forward operator is formed similarly to the ZTD operator (following da_integrat_dz module of GPSPW operator):

$$PW(i,j) = \sum_{k=kts}^{k=kte} (\rho(i,j,k)q(i,j,k)\Delta h), \tag{3}$$

where $\rho$ is an air density.

## 3.2 GNSS data preprocessing

Two kinds of GNSS data are accepted by WRFDA package: ZTD and PW. In order to prepare the GNSS estimates for
GPSPW, a preprocessing is required. The ZTD data is processed according to the following steps:

1. Calculation of the GNSS ZTD using Bernese software for all the stations.
2. Assimilation of the GNSS ZTD obtained in step 1) using the 3DVAR scheme.
3. Calculation of average 'background' corrections from WRF model for each station to reduce the systematic error between WRF and GNSS data and subtracting the corrections from the GNSS ZTD obtained in the step 1)

In the first step, we adjust the formal errors of GNSS ZTD by multiplying them by a factor of 10.5 mm, which is the standard deviation of differences between WRF ZTD and GNSS ZTD according to Dymarska et al., (2017). Next, we remove the observations, which errors exceed 20 mm, which is the standard procedure in GNSS data assimilation (Bennitt and Jupp, 2012).

In the second step, the GNSS data is assimilated in the 3DVAR procedure in order to calculate the corrections that come
from the 'background', which is the WRF model. The corrections for each day are expressed as $O - B$, where $O$ is the 'observation' ZTD (in this case same as $ZTD_{GNSS}$ from the first step), $B$ is 'background' ZTD, i.e. the WRF ZTD. The corrections are then averaged over the entire considered period to obtain one value $(O - B)_{av}$ for each station.

In the third step, the corrected ZTDs are calculated as:

$$ZTD_{corr} = ZTD_{GNSS} - (O - B)_{av} \tag{4}$$

The PW data is processed in a similar way:

1. Calculation of the GNSS PW from GNSS data.
2. Assimilation of the GNSS PW obtained in the step 1) using the 3DVAR scheme

3. Calculation of 'background' corrections and subtracting them from the GNSS PW obtained in the step 1)

From GNSS processing, we can only estimate ZTDs. The PWs in step 1) are calculated in a standard way from GNSS and WRF data as:

$$PW = Q \cdot (ZTD_{GNSS} - ZHD_{WRF}) \tag{5}$$

where $ZHD_{WRF}$ is the hydrostatic delay calculated using Saastamoinen, (1972) formula from pressure from WRF model $p_{WRF}$, height $h$ and latitude $\varphi$ of a GNSS station:

$$ZHD = \frac{0.0022767 p_{WRF}}{1 - 0.00266 \cos(2\varphi) - 0.00000029\, h} \tag{6}$$

The proportionality factor Q is calculated as:

$$Q = \frac{10^6}{R_w (k_3/T_m + k_2')} \tag{7}$$

where $R_w = 461.525$ [J /(K kg)] is the gas constant of a wet air, $k_2' = 22.9726$ [K/hPa] and $k_3 = 375463$ [K$^2$/hPa] are the 'best average' refractivity constants from (Rueger, 2002) and $T_m$ is the mean temperature calculated from $T_{WRF}$ as:

$$T_m = 70.2 + 0.72 \cdot T_{WRF} \tag{8}$$

After calculation of GNSS PW, the processing in steps 2) end 3) is analogical to GNSS ZTD.

## 4 Case studies

All cases presented in this study are selected from the period of May – June, 2013 and location (Central Europe) covering the benchmark campaign of COST Action ES1206 (Douša et al., 2016). Following experiments are considered (Fig. 3): 1) assimilation of ZTD or PW for whole May and June 2013, 2) assimilation of ZTD or PW and ZTD or PW with support of RS and SYNOP for 5-23 May, 2013, 3) case studies a, b and c, showing impact of assimilation of ZTD or PW in severe weather cases which took place during May and June 2013. The mean and standard deviation of ZTD from GNSS and from

WRF are presented in the figure 3. The overall agreement between GNSS and WRF traces are high, however WRF model is negatively biased with respect to the observations and shows less variations. Moreover few cases of significant departure of WRF ZTD from GNSS ZTD is visible in June, two are collocated with case b and case c investigated in this study.

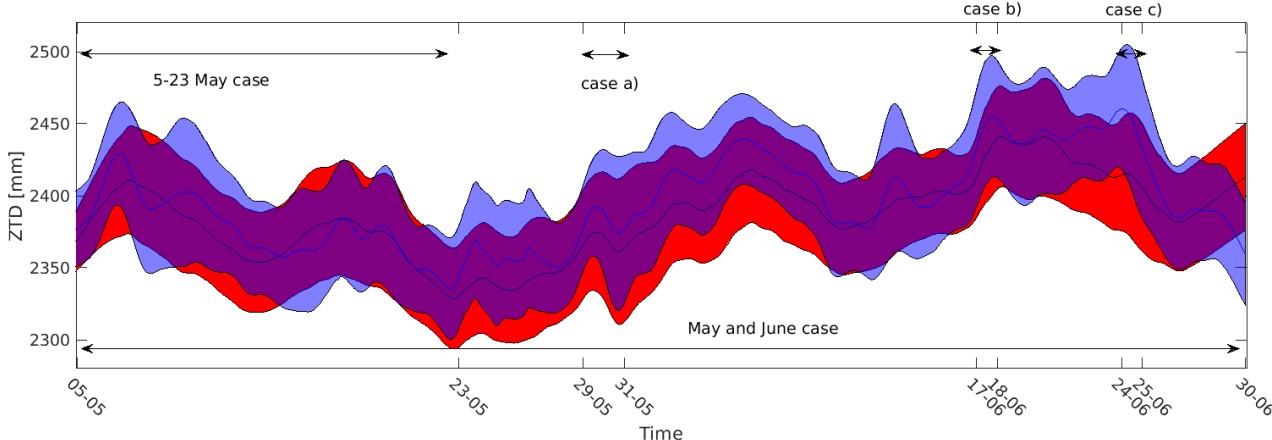

Fig. 3 Time evolution of mean ZTD in the study domain for WRF (red) and GNSS (blue) data (solid lines). Filled area marks standard deviation spread around mean values. Arrows represents time and duration of analyzed cases.

According to synoptic analysis presented in (Douša et al., 2016), the beginning of May 2013 was characterized with cyclonic field over 500hPa, which in turn resulted in precipitation and convection development, moving from west to east. The mid-May weather in the region was developing under the influence of upper level cyclone (500hPa) that brought the cold advection from west. Towards the end of May, series of Atlantic cyclones approached Europe. The end of the month brought a stop to the advection of cold air by upper east ridge, which pushed the cyclones more to the south and brought humid and

warm air to the central Europe. In June, three flooding events were recorded in Czech Republic, which were associated with baroclinic instability developing over area of interest, with a first one (June, 1-3) event unexpected and of disastrous nature while two latter (June, 9-11; June, 23-26) less severe and better predicted. As in this work, we use Poland as a study region, there is a time shift between the events recorded in Czech Republic (described by Douša et al., 2016) and Poland and also the precipitation effects were not as disastrous.

The first severe weather case study (case a) was observed in May, 29 -31, 2013. The weather event is related to an unusual, low-pressure regions: 1) developing over Hungary and moving towards Czechia, 2) developing over the Moldavia and moving towards east of Poland. In these two lows, in the presence of stratified clouds, the cumulonimbus clouds develop and form a supercell. It brought intensive rain and hail, however the precise location of such supercells is not easy to predict (www.meteo.pl).

The second analyzed case (case b) occurred in June, 17 – 18, 2013 and is related to two weather systems: 1) high pressure system with a center in Belarus affecting northern part of Poland, 2) low pressure system over the Bay of Biscay. The cold weather is observed in the north (20° C) and hot and humid in the south (above 30° C). The thermic contrasts and warm unstable air result in occurrence of convective cells located southeast to the region. These cells merged in the late afternoon and formed a supercell storm that moved southward to the Moravy region (www.meteo.pl).

The third case analyzed was June, 24 – 25, 2013. The weather in Europe was driven by high pressure system located over the Atlantic Ocean, as well as large and shallow trough extending to the north to Norway from a weak low centered over the northern part of the Adriatic Sea (with the atmospheric pressure of around 1010 hPa). Secondary cyclogenesis is organized over central Poland in the form of thermal asymmetric low pressure system. A quasi-stationary anabatic cold front spread along this trough changing very slowly its position and bringing cold air from the north (in the western part of Poland), and warm, humid and unstable air masses from the south (in the eastern part of Poland). These conditions are prone to develop strong precipitation, thunderstorms and hail in the central Poland ([www.meteo.pl](www.meteo.pl)).

## 4.1 Assimilation of GNSS observations

The full period of two months is used as a first approach to validate the impact of PW or ZTD data on weather forecasts using radiosonde, GNSS and SYNOP observations

**Comparison to radiosonde profiles**

It is expected that the ZTD and PW assimilation in the first place will affect the 3-D distribution of humidity and temperature. This is summarized in Fig. 4. and Table 2. In the case of relative humidity, assimilation of ZTD significantly improves the model performance in the layer from 2.5 to 10 km, with 22% improvements over the base simulation. At the same time, this assimilation increases the model error for air temperature, but only in the range 2.5 – 5.0km and above 10km. Assimilation of PW has small impact on both relative humidity and air temperature, and the errors, both in terms of the value and vertical distribution is very similar if PW and BASE runs are compared (Fig 4 and Table 2). Only, in the 2.5 – 5.0 km section of troposphere, the assimilation of PW result in significant increase of errors (by 20%).

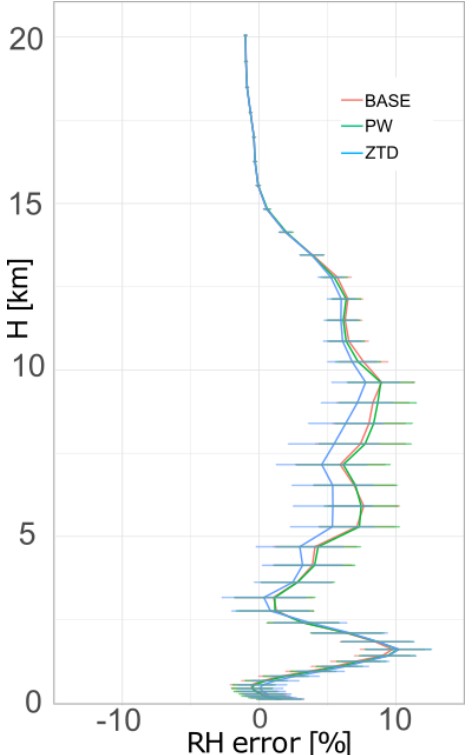
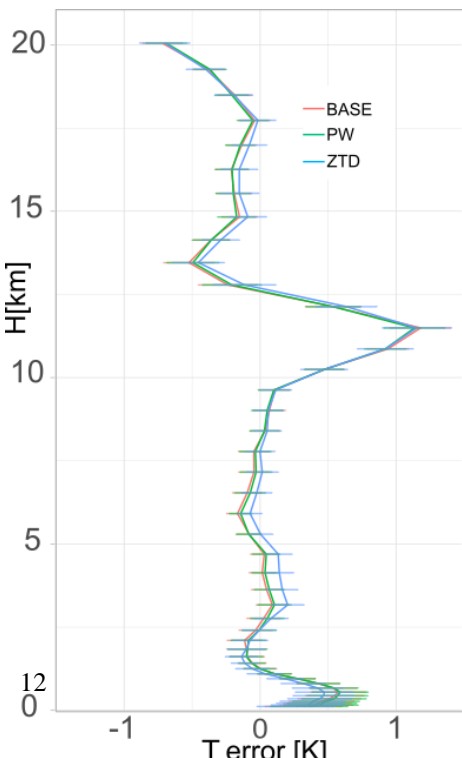

Fig. 4 Vertical distribution of mean error for the BASE, PW and ZTD for May and June 2013 for relative humidity (left panel) and air temperature (right panel). The errors are calculated for the lead time 12h.

**Comparison to GNSS observations**

The assimilation of the GNSS products should also have an impact on the PW and ZTD estimates from the WRF model. We compare the PW and ZTDs calculated from WRF using formulas (2) and (3) respectively using NCAR Command Language (NCL)(NCAR, 2015), before the assimilation ('base') and after the assimilation of both PW and ZTDs. The comparisons are performed for the GNSS data before the processing described in Section 3.1. Thus, the presented biases for the base run are removed from the GNSS to better fit the observations to the model. Station by station comparison (not shown) produces similar results across all stations, hence mean statistics is calculated (Table 3).

In general, the assimilation of the GNSS does not bring a huge improvement in the WRF estimates. For PW, the bias, averaged from all stations equals to 2.6 mm for the base run, and is slightly improved by 0.1mm the PW assimilation and remains same for ZTD assimilation. The average standard deviations for the estimates after assimilation improves by 0.2 mm for both PW and ZTDs. For the ZTD assimilation, there is degradation of the ZTD biases after the assimilation of PW (by 0.5mm) but also improvement in case of ZTD assimilation (by 0.2mm). The assimilation of both PW and ZTD brings an improvement of the ZTD standard deviations for almost all of the stations, therefore the average standard deviations decrease for both PW and ZTD assimilation.

**Comparison to SYNOP**

The further forecast verification is based on a 66 SYNOP stations distributed evenly across Poland (Fig. 2). Table 4 summarizes model forecast performance using: ME, RMSE, corr and IOA. The statistics are calculated for lead times from 1 to 24h for the following parameters: rain intensity (rain), wind speed (wspd), relative humidity (rh2) and temperature (T2).

The overall accuracy of the rain forecast is low, i.e. the base run prediction correlate with the observations in less than 15%, while the same parameter for wind speed is close to 60%, whereas corr for relative humidity is 82% and for temperature 95%. As the assimilation changes the initial conditions of parameters directly linked with the adjoint operator (a transpose of forward operator), the impact while using ZTD should be visible in: pressure $p$ (Eq. 2) (and thus also wind speed), specific humidity $q$ (and thus relative humidity) and temperature $t$. Whereas PW should have impact mostly on specific humidity $q$ (Eq. 3) and thus on rh2 parameter. Rain as a parameter linked with physical parameterization and many other variables such as humidity, vertical and horizontal motion, temperature profile is also sensitive to the GNSS data assimilation.

The results (Table 4) confirm that the assimilation of PW over the whole period of two months affects the forecasts only slightly, the assimilation increases the relative humidity scatter and has negative or neutral impact on the rain ME and RMSE, neutral or positive impact on wind speed and negative or neutral on temperature. Similarly, there is no gain for rainfall forecasts if ZTD is assimilated for the entire period. It has negative impact on wind speed, but there are considerable improvements for relative humidity forecast (15% reduction of ME).

The negative impact of ZTD on wind speed forecast could be linked to the representation of ZHD as a parameter related only to ground based observations of temperature and pressure (Eq. 2), whereas in reality the ZHD is an integral of pressure and temperature across the whole troposphere (Vedel and Huang, 2004).

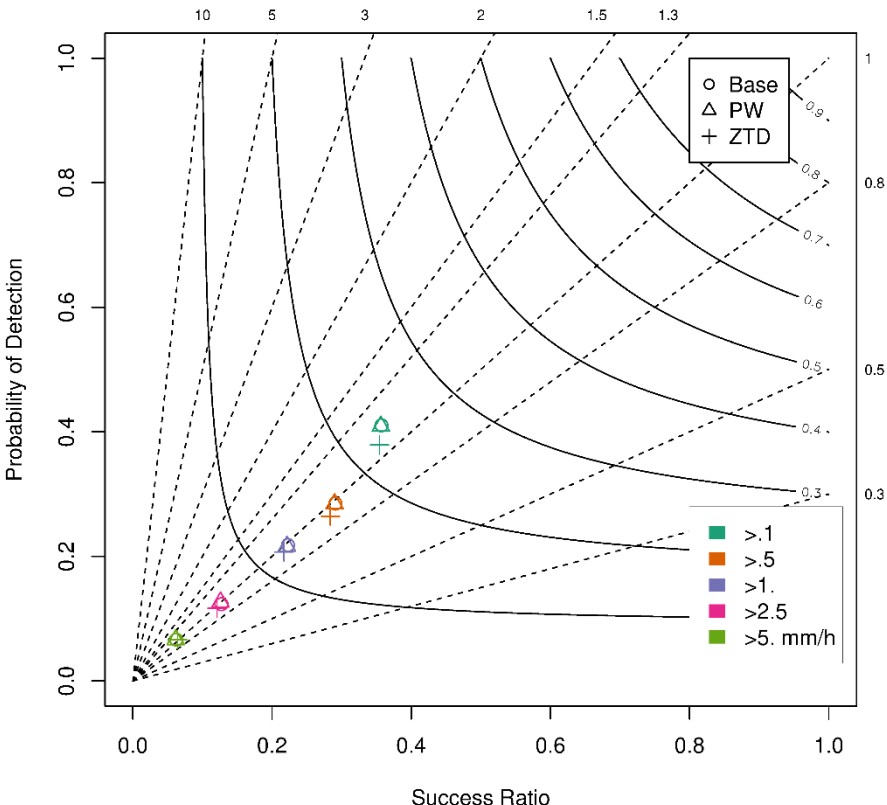

Fig. 5 Performance diagram for assimilation of ZTD and PW for May and June 2013 (lead time <24h). The various colors represent rain intensity classes, whereas the shapes represent datasets.

If the rainfall forecasts are analyzed more closely using the binary verification with data stratification according to rainfall intensity (Fig. 5), it is clear that the PW run is very similar to the base run, regardless the rainfall intensity. The ZTD assimilation leads to overall decrease of the probability of detection.

**4.2 Assimilation of GNSS, RS and SYNOP observations**

In the second experiment, we focus on a short time span covering May with moderate precipitation and standard cyclonic weather, as oppose to the June with the occurrence of major severe weather events (analyzed in 4.3). This experiment is prepared to assess the impact of using RS and SYNOP together with GNSS data in 4DVAR assimilation.

**Comparison to radiosonde profiles**

Comparison to radiosonde data (Fig. 6, Table 5) shows that largest impact on RH is visible in relatively high levels of troposphere 7 – 10 km, however improvements to this parameter are also present for 2.5 – 5.0 km range. Assimilation of PW+SYNOP+RS result in highest gain of quality (in the section 5 – 10 km by 10%). In the lower part this impact is less visible (for any observation type), however below 2.5 km assimilation of PW and ZTD alone increase ME (8% for PW and 2% for ZTD). The impact on temperature is noted only in the 12.5 – 15 km sector. Use of SYNOP and RS data improves forecast in all cases but largest increase is visible in the PW observations and in the 2.5 – 10 km section.

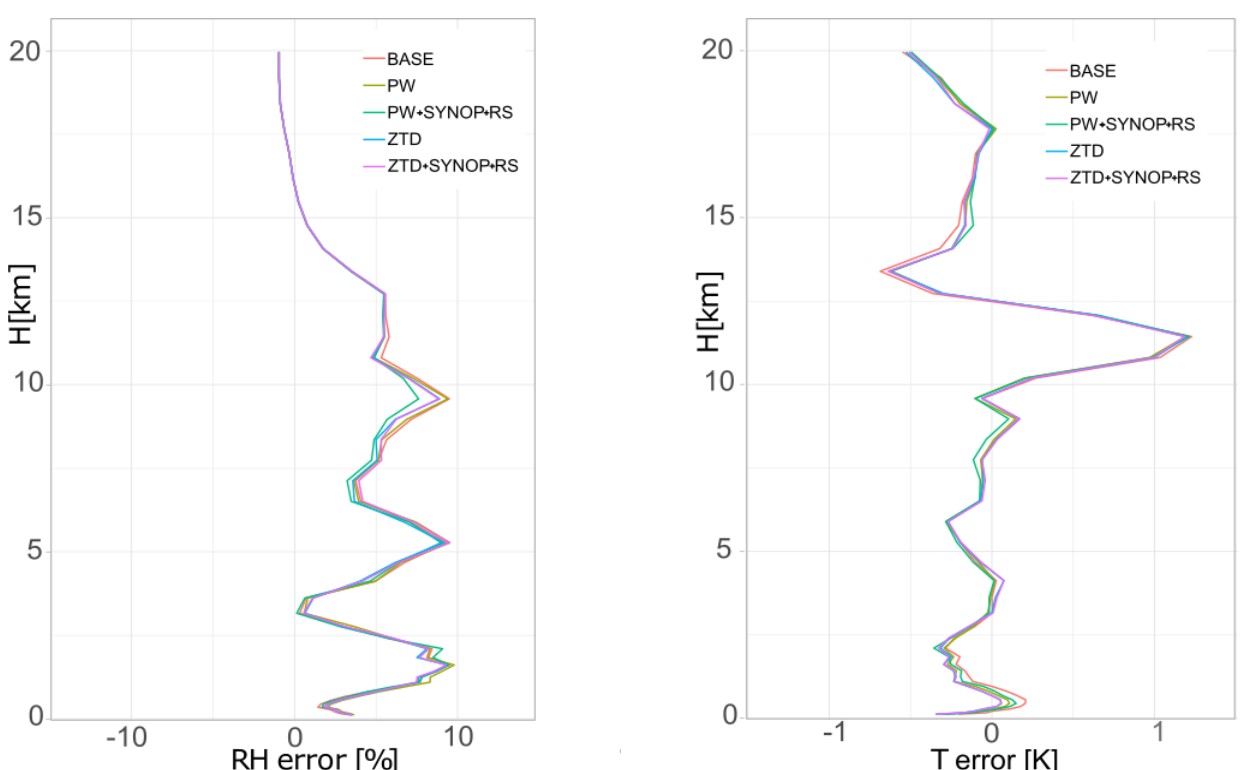

Fig. 6 Vertical distribution of mean error for the BASE, PW, PW+SYNOP+RS, ZTD and ZTD+SYNOP+RS for May 5-23 for relative humidity (left panel) and air temperature (right panel). The errors are calculated for the lead time 12h.

**Comparison to GNSS observations**

Comparing all active GNSS stations ZTDs and PWs to the WRF based ZTDs and PWs (Table 6), one notice no impact on the PW for ZTD and PW assimilation (both bias and std), whereas for the same experiments ZTD bias increased but scatter decreased considerably (reduction of std by 1.2mm). The combination of PW+SYNOP+RS and ZTD+SYNOP+RS if PW field is considered no difference to the base run is noticed, whereas in case of ZTD used as a diagnostic variable, the PW+SYNOP+RS provides best solution, from all assimilation cases, in terms of bias (only by 0.1 mm). Also mean station deviation is lower for assimilated cases than for base simulation.

**Comparison to SYNOP**

The MEs for base run forecasts in May (Table 7) are lower than for May and June, e.g. rain ME is approx. -0.5 whereas May and June is approx. -0.7, May relative humidity ME is approx. 0.4% and May and June is approx. -2.1%. Wind speed errors are similar or slightly higher in May than in May and June. Similar statement is correct for temperature errors. The overall correlation between observations and forecasts is in range from 13% to over 17% for rain, 56% for wspd, 82% for rh2 and

91% for T2, which is a few percent lower than in May and June run (except for rain).

Table 7 shows comparison to SYNOP stations and it confirms that the assimilation of either PW or ZTD has a negative impact on rain forecast in terms of mean error, however positive on all other statistics. The largest improvements are for relative humidity, where all the statistics, except the ME, are improved if compared to base. Surprisingly, adding more observations i.e. SYNOP and RS data does not improve rain or relative humidity forecast in case of ZTD assimilation, but

rather decreases the forecast's quality. Assimilation of both PW and ZTD deteriorates the model performance for wind speed. There is small but positive impact on the T2 forecast in terms of correlation coefficient, but, similarly to RH, mean errors are increased if compared to base run.

As two forecasted parameters are improved: relative humidity and rain (see Table 7), we investigate the lead time differences between base run and four assimilation setups namely: PW, PW+SYNOP+RS, ZTD, ZTD+SYNOP+RS (Fig. 9).

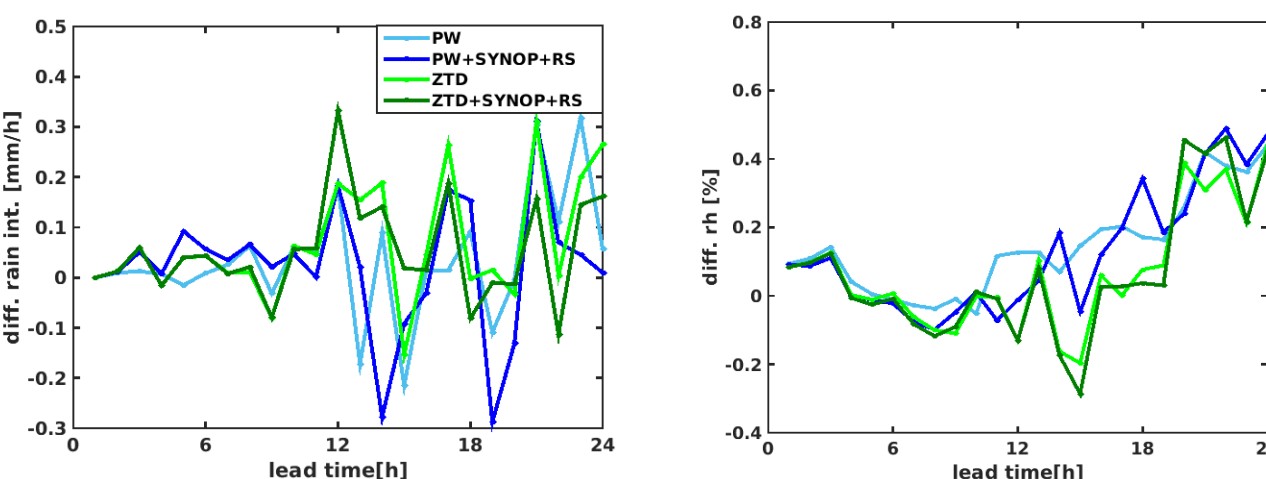

Fig. 7 Performance of rain intensity forecast w.r.t. lead time (positive – improvement with respect to the base run, negative – deterioration with respect to the base run); left panel: ME, right panel: RMSE.

The Fig. 7 ME (left panel) of rain forecast varies significantly during 24 hours, especially in the lead time 10 to 24h and is relatively stable between 1 to 9h (night time). In the scattered section of Figure 9, the ZTD+SYNOP+RS solution seems most of the time positive, while it is negative in the first 9h of forecast. In the first 9 hours of forecast PW+SYNOP+RS reduces the forecast bias. PW and ZTD alone are rarely observed to improve ME of rain forecast. The RMSE pictured on the

right panel of Figure 7 shows similar to ME scattered and compacted sections, however there is clear positive impact of assimilating GNSS observations, especially ZTD+SYNOP+RS in short run (until 24h). Overall, the RS and SYNOP data helps to improve RMSE of rain forecast.

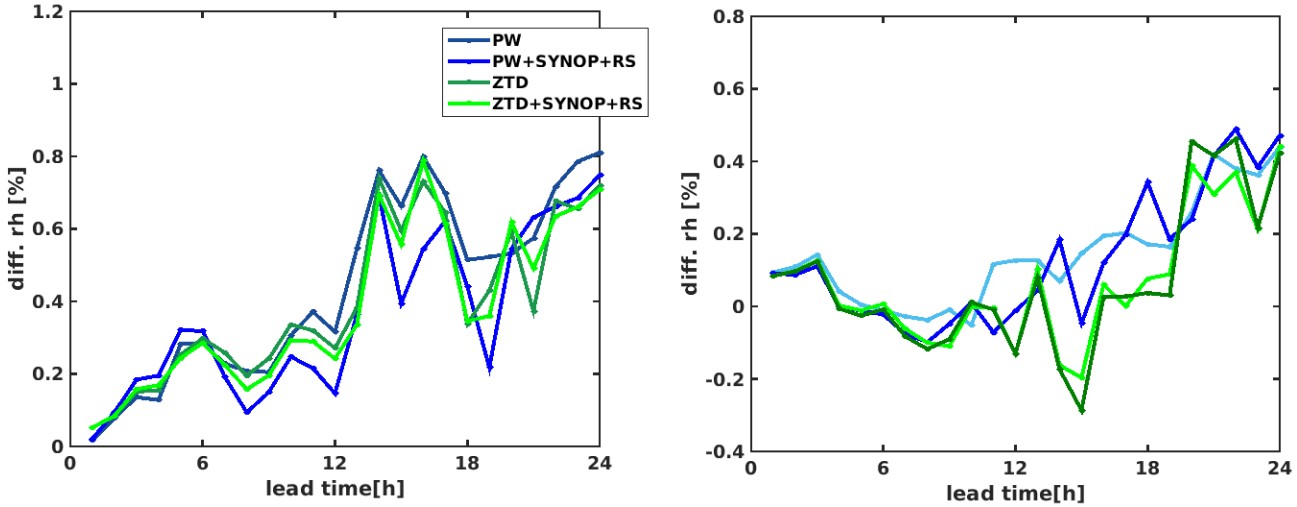

5    Fig.8 Performance of rh2 forecast w.r.t. lead time. Left panel: ME, right panel: RMSE.

Less variation between the four scenarios is observed for relative humidity errors (Fig.8). Both, ME and RMSE are reduced while assimilating each data type, with an exception of lead time 15h, when RMSE increases (more when ZTD is used, less when PW is used). Moreover, the highest reduction of ME is noticed between 12h and 18h of lead time for PW+SYNOP+RS

10   scenario, but other scenarios are also showing positive impact. It is also worth to mention that the rh2 forecast RMSE is reduced after 12h lead time whereas bias is constantly reduced starting form the first hour of forecast.

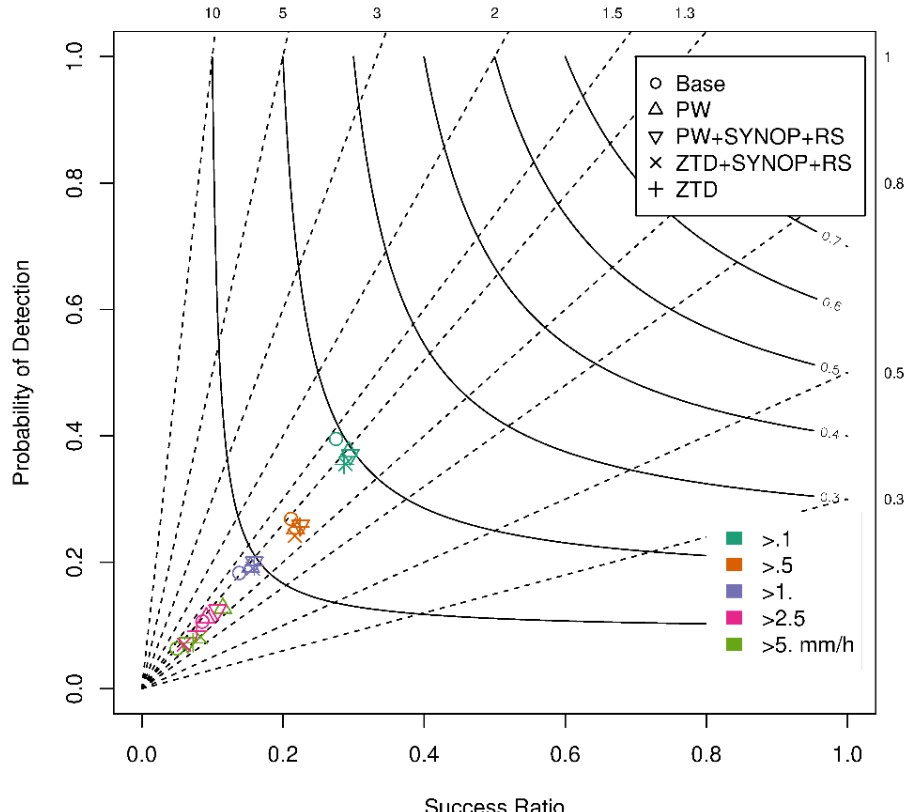

Fig. 9 Performance diagram for assimilation of PW, PW+SYNOP+RS, ZTD+SYNOP+RS, ZTD for May 2013 (lead time <24h). The various colors represent rain intensity classes, whereas the shapes represent datasets.

Binary rain analysis shows that the impact of data assimilation on rainfall forecasts changes with rainfall intensity (Fig. 9). For the rainfall intensity above 0.1 mm/h, there are small improvements for all the model runs, if compared to the base run in terms of Success Ratio, but the Probability of Detection is smaller. The positive impact of data assimilation is much stronger for higher rainfall intensities. For the thresholds exceeding 1.0 mm/h, both Probability of Detection and Success Ratio are
10 improved if compared to the base run. The improvement is especially large for PW data assimilation and threshold >5.0 mm/h.

**4.3 Severe weather cases**

The final test is performed using selected 3 cases with strong instabilities and supercell storms. The overall impact of GNSS data in all cases is similar: if there is any reduction in uncertainty it is visible mostly in rain and relative humidity forecast, with a small negative or neutral impact on the wind speed and temperature forecasts.

**Case a) May, 29-31, 2013**

The comparison of WRF-based RH and T with radiosonde shows that for case a) (Fig. 10) the impact of ZTD assimilation is introducing large change to the initial conditions and forecast. This impact is in RH positive in the first 2 km and between 6 and 10 km and might be negative close to 2.5 km. Temperature show mixed results with stronger influence of ZTD and smaller for PW but across whole profile this impact is small but positive. Moreover ZTD increase agreement between model

and RS in the bottom 2 km of temperature profile.

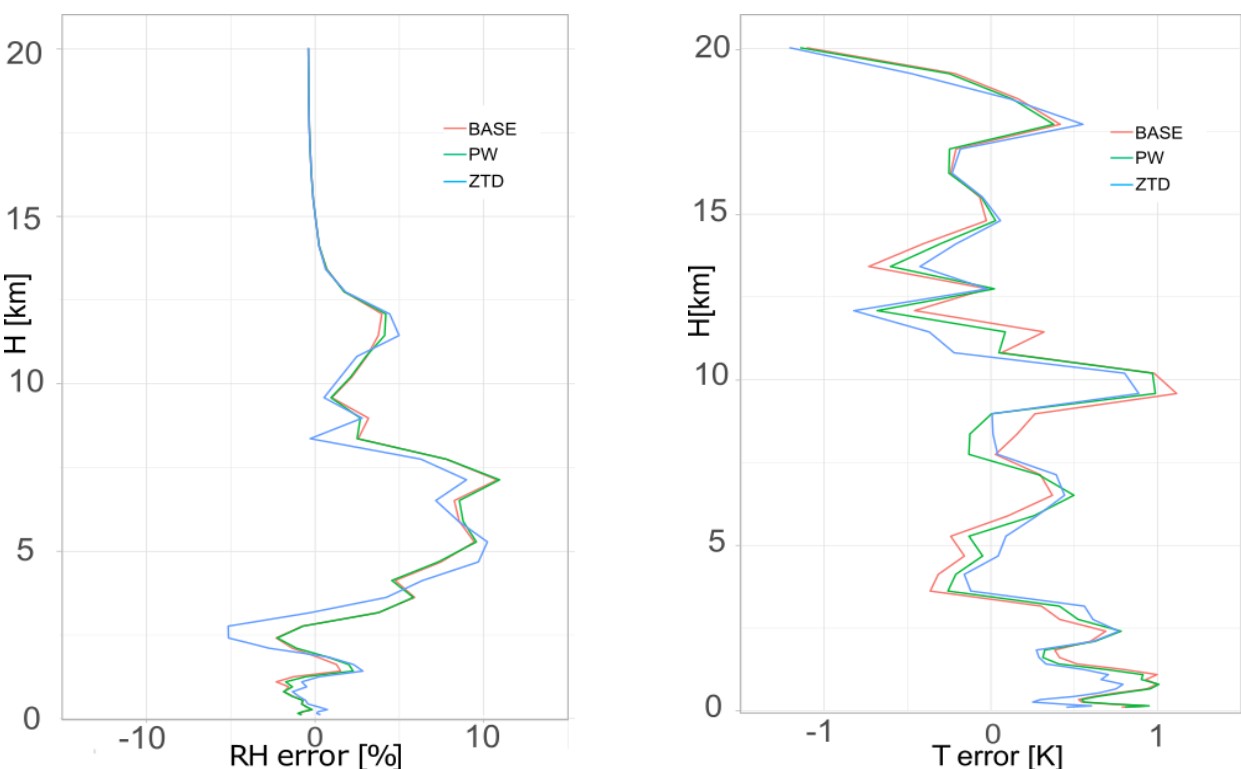

Fig.10 Vertical distribution of mean error for the BASE, PW, and ZTD for case a) in May, 29 -31, 2013 for relative humidity (left panel) and air temperature (right panel). The errors are calculated for the lead time 12h.

Comparing forecast with assimilation of ZTD and PW with GNSS observations (Table 8), shows that both data reduce scatter of the observations but ZTD is also reducing systematic effects, while PW is increasing bias.

According to Table 9. presenting comparison to SYNOP data mixed results are observed for case a). Rain forecast shows better performance if PW is compared to base run. Assimilation of ZTD for this case deteriorates model performance except

for mean error. Humidity forecast (rh2) is improved in terms of RMSE and corr, when ZTD is assimilated. For PW assimilation, model performance is worse if compared to Base.

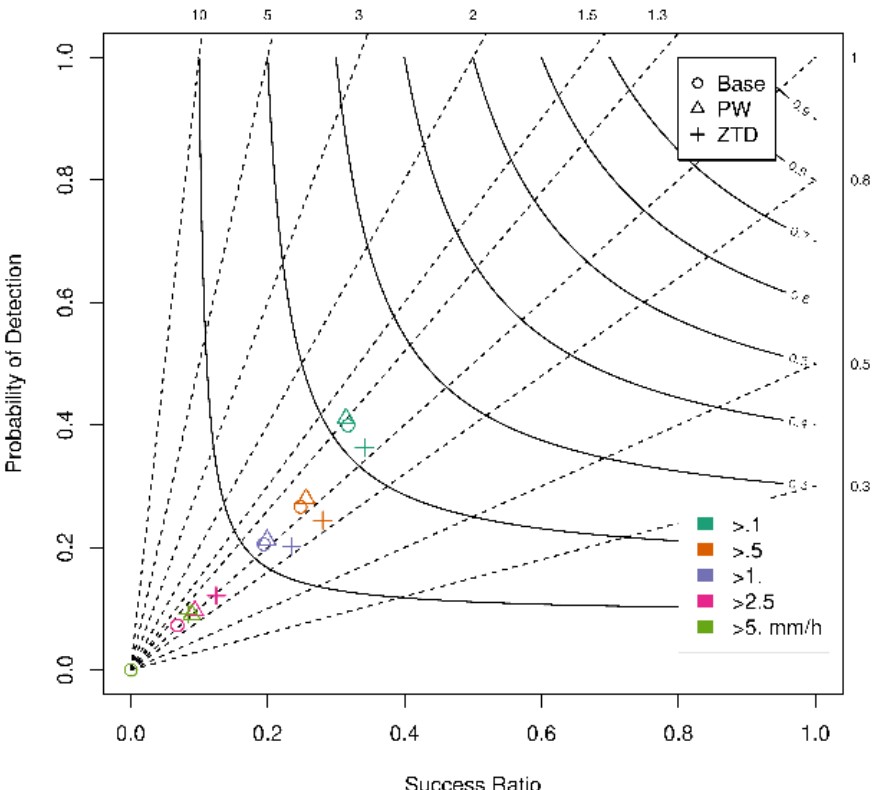

Fig. 11 Performance diagram for assimilation of PW, ZTD for case a) (lead time <24h) ). The various colors represent rain intensity classes, whereas the shapes represent datasets.

Binary analysis depicted on Fig. 11 shows the positive impact of assimilating of PW in a rainfall rate above 0.5 mm/h, and ZTD above 2.5 mm/h.

**Case b) June, 17 – 18, 2013**

The comparison of WRF-based RH and T with radiosonde shows that for case b) (Fig. 12) the impact of ZTD assimilation is introducing large change to the initial conditions and forecast, while PW has almost no impact. The ZTD impact is in RH negative in the first 5 km and positive from 7 to 10km. Temperature profile shows positive results with stronger influence of ZTD and almost none for PW. It is visibly positive in the bottom part of the troposphere (first 1 km).

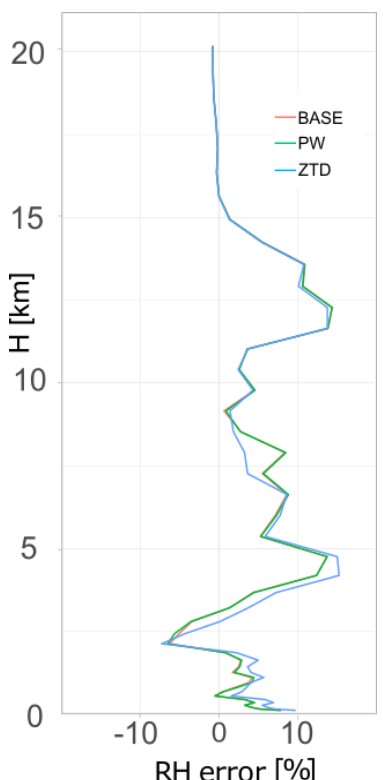 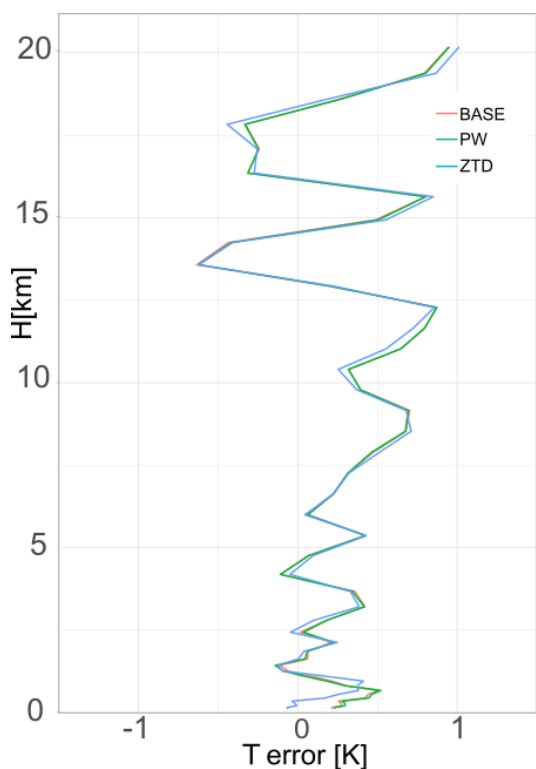

Fig.12 Vertical distribution of mean error for the BASE, PW, and ZTD for case b) June, 17 – 18, for relative humidity (left panel) and air temperature (right panel). The errors are calculated for the lead time 12h.

Comparing forecast with assimilation of ZTD and PW with GNSS observations (Table 8), shows that both data reduce
scatter of the observations. The bias for PW parameter slightly increase for both type of assimilated observations, however ZTD bias is reduced.

Comparison to SYNOP data (Table 9) shows that, the assimilation of PW does not change model performance for rainfall, except for ME reduction. The model performance for other parameters is similar or slightly worse than base. ZTD assimilation has small positive impact on RMSE, corr and IOA of rain intensity forecast, but negative on ME. Relative
humidity ME is reduced by assimilation of ZTD by 43%, all other measures are better for base run. In the local type of rain in SE Poland, as in this case, it is impossible to present statistically sound results for 5 rainfall classes, hence we did not provide binary rain analysis for this case.

**Case c) June, 24 – 25, 2013**

The comparison of WRF-based RH and T with radiosonde shows that for case c) (Fig. 13 ) the impact of ZTD assimilation is
introducing large change to the initial conditions and consequently to the forecast. This impact is in RH positive in the first 1 km and negative close to 2.5 km and clearly positive at 5 km. Temperature show mixed results, however it is worth to point

out that from 2.5 km to 5km there is positive bias introduced to the forecast by both PW and ZTD. Temperature is improved once ZTD is assimilated in the bottom part of troposphere (first 1 km).

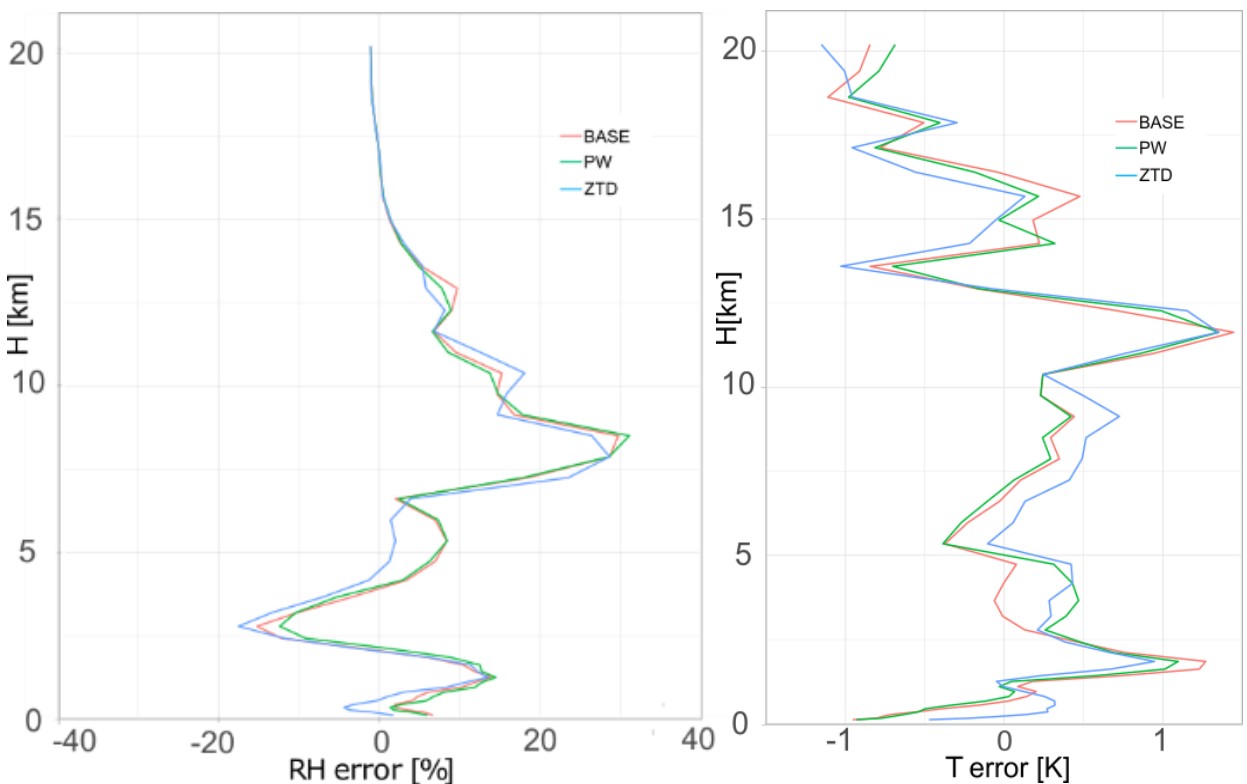

Fig. 13 Vertical distribution of mean error for the BASE, PW and for case c) June, 24-26 for relative humidity (left panel) and air temperature (right panel). The errors are calculated for the lead time 12h

Comparing forecast with assimilation of ZTD and PW with GNSS observations (Table 8), shows that both data reduce scatter of the observations. The bias for PW parameter slightly increase (ZTD assimilation) or is neutral (PW assimilation). The bias for ZTD decrease for PW assimilation and slightly increase for ZTD assimilation.

Third case once compared to the SYNOP shows also small but positive impact of ZTD and, especially PW data assimilation on rainfall forecasts, except for mean error. Errors statistics are improved also for wind speed. In the case of PW run, there is also a gain for relative humidity, while for ZTD error statistics are worse if compared to the Base run.

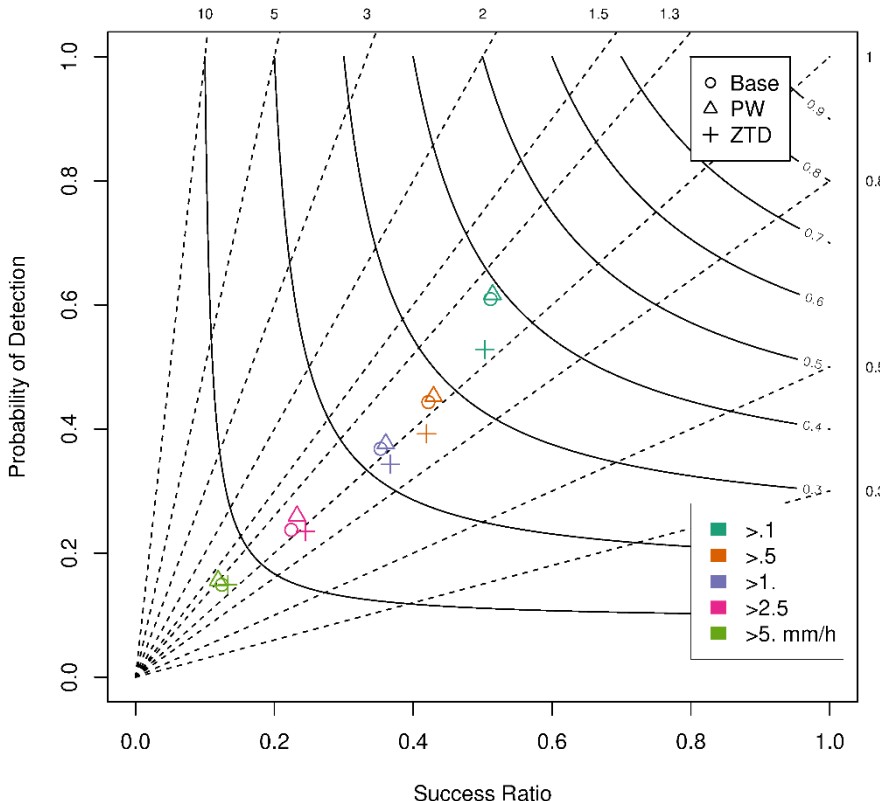

Fig. 14 Performance diagram for assimilation of PW, ZTD for case c). The various colors represent rain intensity classes, whereas the shapes represent datasets.

In the performance diagram in Fig. 14, the rain rate forecasts are improved with PW w.r.t. the base forecast, but worse when ZTD is assimilated. This effect is visible for all rain rates lower than 1mm/h and this discrepancy disappears for rain rates in the 2.5 mm/h class, where both ZTD and PW have positive impact, whereas no impact is noticed for rainfall rates above 5 mm/h.

10 **5 Summary and conclusions**

In this study, we have analyzed 2 months (May and June 2013) of 4DVAR assimilation of GNSS ground-based observations in WRF model, from over 100 stations in Poland. Two major approaches were investigated using GPSPW operator: assimilation of PW and ZTD. For shorter time period of 18 days in May additional data were assimilated, namely: RS and SYNOP observations across Poland. Moreover, three different case studies related to severe weather occurrence were 15 investigated. All were linked to a supercell development and intense rain.

The major conclusion of this study is that for analyzed time period with more than 100 stations involved in the experiment, there are two parameters significantly changed once GNSS data are assimilated in the WRF model using GPSPW operator and these are: moisture field and rain. Other parameters such as pressure or temperature field are not changing initial conditions significantly. The GNSS observations improves forecast in the first 24 hours but with strongest impact starting from 9h lead time. It is worth notice that even moderate quality NRT estimates used in this study (ZTD discrepancy ~10-15mm) are improving relative humidity forecast, moreover the impact of ZTD is in the vertical profile positive (over 20% decrease of mean error) starting from 2.5 km upward. Even if the humidity forecast in lower part of troposphere (below 2.5km) after GNSS data assimilation deteriorates, the SYNOP observations confirms that ZTD has positive impact on the rh2 parameter. Assimilation of PW has less significant impact on both humidity and rain forecasts in the vertical profile and on the ground, it improves slightly rain forecast and deteriorates by few percent the humidity prediction. The best results for use of integrated water vapor information is produce by adding radiosonde and SYNOP data to the assimilation system as it improves distribution of humidity in the vertical profile. In analyzed three severe weather cases PW always improved rain forecast and ZTD was always reducing humidity field bias. Binary rain analysis shows that GNSS parameters have significant impact of rain forecast in the class above 1mm/h, while adding more observations such as radiosonde and SYNOP result in larger increase of quality. In each case however the impact of GNSS data was different, therefore below we provide a brief summary for each case.

May and June case results showed that in the lead time of 24h, the assimilation of GNSS data (both ZTD and PW) had varying impact on a number of parameters. The assimilation of PW over whole period of two months: marginally diverted the humidity and temperature 3D performance, which is seen on the radiosonde profiles. It also improves by 0.1mm the agreement between WRF model based and GNSS based PW observations. Assimilating PW slightly increased correlation of daily rainfall rates, increased relative humidity scatter and had positive impact on the rain RMSE, neutral impact on wind speed and negative or neutral on temperature. However, the binary analysis of rain rate in five intensity classes revealed that the forecasts with assimilation of PW improves forecasts scores in high intensity rain above 2.5 mm/h.

Assimilation of ZTD had a large impact on the vertical profile of both temperature and humidity, as retrieved from comparison to radiosonde data especially in the range 5 to 10 km. It reduced by 0.1 mm bias and by 0.5 mm std between base run and assimilation run for WRF derived ZTD and GNSS derived ZTD. Assimilation of ZTD reduced the ME for humidity by 16%, while it had slight negative impact on rain, temperature and wind speed bias. The binary analysis of rain rate in five intensity classes revealed that the forecasts with assimilation of ZTD improves forecasts scores only in the highest rain rate class.

More detailed study focused on May, 5-23, (with non-severe weather events) showed that the assimilation of either PW or ZTD reduced mean error in the humidity forecast in the vertical direction, most successfully for PW+SYNOP+RS data. It had positive impact on rain forecast and relative humidity forecast. The assimilation of PW improved rain rate RMSE by 5% and had negative impact on bias (7% increase). The relative humidity forecast bias was doubled with assimilation however RMSE was reduced by minimum 1%. The assimilation of ZTD improved rain rate RMSE by 6% and had negative impact on

bias (14% increase). The relative humidity forecast bias was doubled with assimilation, RMSE was reduced by less than 1%. Adding SYNOP stations and radiosonde did not bring any further improvements in forecasting humidity or rain but reduced the errors in wind speed and temperature data. Furthermore, the analysis of lead time w.r.t. the errors revealed that for rain rate ME error varies in time (both negative and positive impacts are present), whereas the RMSE for data with assimilation is

considerably improving with time. In case of relative humidity both ME and RMSE are reduced when GNSS data are assimilated, largest gain in quality is observed for PW+SYNOP+RS data set. The binary analysis show positive impact of GNSS data assimilation especially for rain rates above 1 mm/h..

In the analyzed severe rain cases, the assimilation of GNSS in case a), b) and c) brings reduction of ME and RMSE or at least RMSE for key sensitive parameters such as rain rate, relative humidity. Binary rain rate forecast performance analysis

shows that the intensive rain is better predicted once GNSS data are assimilated. Further research, based on larger number of cases, is required to investigate what are the reasons of different impact of GNSS data on model forecast.

**Author contribution**

Witold Rohm - provided leadership as project PI, wrote the manuscript, prepared art works and tables, coordinated research. Jakub Guzikowski – run the WRF simulations and assimilation.

Karina Wilgan – prepared the GNSS data and the bias corrections, wrote the GNSS preprocessing section and reviewed the manuscript.

Maciej Kryza – performed the verification of the simulations, wrote the model evaluation section and reviewed the manuscript.

**Competing interests**

Authors confirm no conflict of interest

**Data**

GNSS data used in this study are available from the Institute of Geodesy and Geoinformatics data base MaGDA, access could be granted to any individual after mailing to jan.sierny@igig.up.wroc.pl. Meteorological data were provided by the

Institute of Meteorology and Water Management – National Research Institute. National Centers for Environmental Prediction/National Weather Service/NOAA/U.S. Department of Commerce, 2000: NCEP FNL Operational Model Global Tropospheric Analyses as boundary and initial conditions for the model run.

**Acknowledgements:**

Authors would like to acknowledge input from Dr. Marek Błaś in analysing weather conditions discussed in the manuscript. This work has been supported by National Science Centre project No UMO-2013/11/D/ST10/03473, COST Action ES1206 GNSS4SWEC (www.gnss4swec.knmi.nl), and the Wroclaw Center of Networking and Supercomputing (http://www.wcss.wroc.pl/) computational Grant using MATLAB Software License no: 101979 and computational grant no: 170. Meteorological data were provided by the Institute of Meteorology and Water Management – National Research Institute. National Centers for Environmental Prediction/National Weather Service/NOAA/U.S. Department of Commerce, 2000: NCEP FNL Operational Model Global Tropospheric Analyses, continuing from July 1999. Research Data Archive at the National Center for Atmospheric Research, Computational and Information Systems Laboratory, Boulder, CO. [Available online at  http://dx.doi.org/10.5065/D6M043C6.]

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

Table 1. The WRF configuration used in the experiment

| Parameters | Domain 1 | Domain 2 |
|---|---|---|
| spatial resolution | 12 km x 12 km | 4 km x 4 km |
| vertical levels | 48 | 48 |
| microphysics | Thompson (Thompson et al., 2004) | Thompson (Thompson et al., 2004) |
| cumulus | Kain-Fritsch (Kain, 2004) | - |
| longwave radiation | RRTM (Mlawer, 1997) | RRTM (Mlawer, 1997) |
| shortwave radiation | Dudhia (Dudhia, 1989) | Dudhia (Dudhia, 1989) |
| surface layer | MM5 (Paulson, 1970) | MM5 (Paulson, 1970) |
| planetary boundary layer | Yonsei University scheme (Hong et a., 2006) | Yonsei University scheme (Hong et al., 2006) |

Table 2 Mean error for RH and T calculated for selected height classes for May and June 2013. Lead time is 12 h. Colors decode improvement (green), deterioration (red) or no impact (yellow), of forecast with the use of GNSS data.

| Height | RH mean error [%] | | | T mean error [K] | | |
|---|---|---|---|---|---|---|
| | BASE | PW | ZTD | BASE | PW | ZTD |
| <2.5 km | 3.35 | 3.47 | 3.85 | 0.25 | 0.26 | 0.19 |
| 2.5 – 5.0 km | 2.26 | 2.74 | 2.01 | 0.04 | 0.06 | 0.14 |
| 5.0-10.0 km | 7.55 | 7.71 | 5.94 | -0.03 | -0.02 | 0.02 |
| >10.0 km | 2.32 | 2.25 | 2.14 | 0.00 | 0.00 | 0.04 |

Table 3. Bias and standard deviation for PW and ZTD calculated for all GNSS stations in the experiment for May and June 2013. Lead time < 24 h. Colors decode improvement (green), deterioration (red) or no impact (yellow), of forecast with the use of GNSS data.

| Parameter | run | bias[mm] | std [mm] |
|---|---|---|---|
| PW | Base | 2.6 | 4.9 |
| | PW | 2.5 | 4.7 |
| | ZTD | 2.6 | 4.7 |
| ZTD | Base | -8.3 | 26.5 |
| | PW | -8.8 | 25.7 |
| | ZTD | -8.1 | 26.0 |

Table 4. Impact of assimilation of PW and ZTD using 4DVAR operators, validated against SYNOP observations, for June and May (lead time <24h). Colors decode improvement (green), deterioration (red) or no impact (yellow), of forecast with the use of GNSS data.

| run | rain | | | | wspd | | | | rh2 | | | | T2 | | | |
|---|---|---|---|---|---|---|---|---|---|---|---|---|---|---|---|---|
| | me | rmse | corr | IOA | me | rmse | corr | IOA | me | rmse | corr | IOA | me | rmse | corr | IOA |
| Base May and June | -0.680 | 2.567 | 0.123 | 0.380 | 0.055 | 1.522 | 0.589 | 0.760 | -2.098 | 10.643 | 0.823 | 0.903 | -0.351 | 2.244 | 0.918 | 0.956 |
| PW | -0.684 | 2.559 | 0.124 | 0.381 | 0.057 | 1.520 | 0.590 | 0.760 | -2.104 | 10.658 | 0.822 | 0.902 | -0.352 | 2.246 | 0.918 | 0.956 |
| ZTD | -0.720 | 2.570 | 0.122 | 0.380 | 0.083 | 1.526 | 0.584 | 0.756 | -1.765 | 10.577 | 0.823 | 0.904 | -0.409 | 2.243 | 0.919 | 0.956 |

Table 5. Mean error for RH and T calculated for selected height classes for May 5-23 2013. Lead time is 12 h. . Colors decode improvement (green), deterioration (red) or no impact (yellow), of forecast with the use of GNSS data. Colors decode improvement (green), deterioration (red) or no impact (yellow), of forecast with the use of GNSS data.

| Height | RH mean error [%] | | | | | T mean error [K] | | | | |
|---|---|---|---|---|---|---|---|---|---|---|
| | BASE | PW | PW+SYNO P+RS | ZTD | ZTD+SYN OP+RS | BASE | PW | PW+SYNO P+RS | ZTD | ZTD+SYN OP+RS |
| <2.5 km | 5.05 | 5.49 | 5.19 | 5.19 | 5.18 | -0.04 | -0.12 | -0.10 | -0.15 | -0.15 |
| 2.5 – 5.0 km | 3.16 | 3.31 | 2.92 | 2.99 | 3.06 | -0.04 | -0.04 | -0.05 | -0.02 | -0.02 |
| 5.0-10.0 km | 6.47 | 6.43 | 5.72 | 6.05 | 6.35 | -0.07 | -0.08 | -0.10 | -0.07 | -0.06 |
| >10.0 km | 2.15 | 2.05 | 2.01 | 2.04 | 2.03 | 0.01 | 0.02 | 0.03 | 0.02 | 0.02 |

10    Table 6. Bias and standard deviation for PW and ZTD calculated for all GNSS stations in the experiment for May, 5-23, 2013. Lead time < 24 h. Colors decode improvement (green), deterioration (red) or no impact (yellow), of forecast with the use of GNSS data.

| parameter | run | bias[mm] | std [mm] |
|---|---|---|---|
| PW | Base | 6.3 | 5.2 |
| | PW | 6.3 | 5.2 |
| | ZTD | 6.3 | 5.3 |
| | PW+SYNOP+RS | 6.3 | 5.2 |
| | ZTD+SYNOP+RS | 6.3 | 5.2 |
| ZTD | Base | -2.0 | 25.1 |
| | PW | -2.9 | 23.9 |
| | ZTD | -2.8 | 23.9 |
| | PW+SYNOP+RS | -2.1 | 24.6 |
| | ZTD+SYNOP+RS | -2.4 | 24.6 |

Table 7. Impact of assimilation of PW, ZTD, RS and SYNOP using 4DVAR operators, validated against SYNOP observations, for May, 5-23, 2013 (lead time <24h). Colors decode improvement (green), deterioration (red) or no impact (yellow), of forecast with the use of GNSS data.

| run | rain | | | | wspd | | | | rh2 | | | | T2 | | | |
|---|---|---|---|---|---|---|---|---|---|---|---|---|---|---|---|---|
| | me | rmse | corr | IOA | me | rmse | corr | IOA | me | rmse | corr | IOA | me | rmse | corr | IOA |
| Base May | -0.475 | 2.027 | 0.132 | 0.383 | 0.046 | 1.551 | 0.565 | 0.742 | 0.352 | 10.987 | 0.819 | 0.903 | -0.668 | 2.333 | 0.914 | 0.952 |
| PW | -0.509 | 1.933 | 0.171 | 0.419 | 0.080 | 1.563 | 0.557 | 0.737 | 0.771 | 10.855 | 0.825 | 0.906 | -0.743 | 2.332 | 0.916 | 0.952 |
| ZTD | -0.543 | 1.903 | 0.166 | 0.417 | 0.082 | 1.554 | 0.559 | 0.738 | 0.729 | 10.938 | 0.822 | 0.904 | -0.734 | 2.343 | 0.915 | 0.951 |
| PW+ SYNOP+RS | -0.494 | 1.990 | 0.150 | 0.401 | 0.043 | 1.556 | 0.554 | 0.735 | 0.697 | 10.875 | 0.824 | 0.906 | -0.749 | 2.336 | 0.916 | 0.952 |
| ZTD+SYNOP+RS | -0.528 | 1.939 | 0.159 | 0.413 | 0.079 | 1.558 | 0.557 | 0.737 | 0.717 | 10.940 | 0.822 | 0.904 | -0.751 | 2.336 | 0.916 | 0.952 |

Table 8. Bias and standard deviation for PW and ZTD calculated for all GNSS stations in the experiment for selected cases (case a) May, 29-31, 2013; (case b) June, 17-19, 2013; (case c) June, 24 -26, 2013. Lead time < 24 h. Colors decode
10  improvement (green), deterioration (red) or no impact (yellow), of forecast with the use of GNSS data.

| parameter | run | bias[mm] | std [mm] |
|---|---|---|---|
| PW | Base (case a) | -0.6 | 2.9 |
| | PW (case a) | -0.6 | 2.8 |
| | ZTD (case a) | -0.4 | 2.7 |
| | Base (case b) | -0.7 | 2.7 |
| | PW (case b) | -0.8 | 2.4 |
| | ZTD (case b) | -0.8 | 2.4 |
| | Base (case c) | 0.4 | 3.4 |
| | PW (case c) | 0.4 | 3.2 |
| | ZTD (case c) | 0.5 | 3.2 |
| ZTD | Base (case a) | -14.6 | 23.6 |
| | PW (case a) | -15.2 | 23.3 |
| | ZTD (case a) | -14.2 | 23.3 |
| | Base (case b) | -11.0 | 22.0 |
| | PW (case b) | -10.6 | 21.1 |
| | ZTD (case b) | -10.6 | 20.8 |

| | | |
|---|---|---|
| Base (case c) | -34.9 | 23.9 |
| PW (case c) | -34.7 | 23.6 |
| ZTD (case c) | -35.0 | 23.7 |

Table 9. Impact of assimilation of PW, ZTD, RS and SYNOP using 4DVAR operators, validated against SYNOP observations, for selected cases  (case a) May, 29-31, 2013; (case b) June, 17-19, 2013; (case c) June, 24 -26, 2013 (lead time <24h). Colors decode improvement (green), deterioration (red) or no impact (yellow), of forecast with the use of GNSS data.

| run | rain | | | | wspd | | | | rh2 | | | | T2 | | | |
|---|---|---|---|---|---|---|---|---|---|---|---|---|---|---|---|---|
| | me | rmse | corr | IOA | me | rmse | corr | IOA | me | rmse | corr | IOA | me | rmse | corr | IOA |
| Base (case a) | -1.233 | 3.412 | 0.020 | 0.327 | 0.416 | 1.870 | 0.569 | 0.722 | -3.049 | 10.760 | 0.811 | 0.891 | -0.102 | 2.506 | 0.862 | 0.923 |
| PW (case a) | -1.166 | 3.360 | 0.036 | 0.353 | 0.429 | 1.872 | 0.566 | 0.720 | -3.130 | 10.807 | 0.810 | 0.891 | -0.093 | 2.517 | 0.861 | 0.923 |
| ZTD (case a) | -1.155 | 3.609 | -0.006 | 0.313 | 0.463 | 1.916 | 0.555 | 0.710 | -2.774 | 10.623 | 0.811 | 0.893 | -0.169 | 2.484 | 0.864 | 0.924 |
| Base (case b) | -2.016 | 4.626 | -0.082 | 0.331 | 0.152 | 1.324 | 0.466 | 0.680 | -0.597 | 10.197 | 0.826 | 0.907 | -0.501 | 2.155 | 0.930 | 0.957 |
| PW (case b) | -2.012 | 4.626 | -0.082 | 0.330 | 0.155 | 1.326 | 0.467 | 0.681 | -0.566 | 10.244 | 0.824 | 0.906 | -0.510 | 2.159 | 0.929 | 0.957 |
| ZTD (case b) | -2.026 | 4.625 | -0.061 | 0.332 | 0.175 | 1.362 | 0.427 | 0.656 | -0.092 | 10.286 | 0.823 | 0.906 | -0.614 | 2.204 | 0.927 | 0.955 |
| Base (case c) | -0.701 | 3.240 | 0.223 | 0.490 | -0.065 | 1.864 | 0.575 | 0.750 | -4.444 | 10.970 | 0.760 | 0.842 | -0.004 | 2.210 | 0.865 | 0.929 |
| PW (case c) | -0.739 | 3.135 | 0.250 | 0.510 | -0.064 | 1.859 | 0.577 | 0.751 | -4.455 | 10.913 | 0.764 | 0.843 | -0.004 | 2.205 | 0.866 | 0.929 |
| ZTD (case c) | -0.803 | 3.167 | 0.250 | 0.508 | -0.030 | 1.835 | 0.576 | 0.750 | -4.433 | 11.066 | 0.760 | 0.841 | -0.044 | 2.216 | 0.868 | 0.930 |