# Peer review of "4DVAR assimilation of GNSS zenith path delays and precipitable water into a numerical weather prediction model WRF"

_Atmospheric Measurement Techniques, 2018_

## Referee Comment (RC1) · Anonymous Referee #1 · 27 Aug 2018

Reviewers comments

on

4DVAR assimilation of GNSS zenith path delays and precipitable water into a numerical weather prediction model WRF

by

Witold Rohm, Jakub Guzikowski, Karina Wilgan and Maciej Kryza

Journal Atmos. Meas. Tech., https://doi.org/10.5194/amt-2018-263

[Figure]

General comments ─────────────────

This Paper is concerned with evaluating the impact of GNSS observations within the WRF limited area data assimilation and modelling system over an area covering Poland. Previous work in the field of research is throughly reviewed. However, ufortunately, misinterpetations were found.

The Paper has a sound scientific basis in the sense the effect of utilizing GNSS-based observations on the quality of numerical weather prediction is investigated. The experiments carried out are quite clearly described, although some improvements can be made. Unfortunately, in my opinion, the results of the experiments is not well enough evaluated and presented. Over such a small area and for a humidity related observations it is not advisable to evaluate the impact including forecasts up to 48 h and focus on surface observations and precipitation. In my opinion one should focus on forecasts up to roughly +12h and on verification also on upper air fields. In addition the statistical significance of the results should be presented.

Major Revisions of the Paper are needed, in particular regarding the evaluation of the parallel experiment. After these have bee carried out the Paper can be considered for scientific publication in Journal Atmos. Meas. Tech..

Please find below more detailed comments and suggestions are listed.

More Specific Comments ─────────────-

Page 1. line 11: I suggest change 'codified' to 'represented'

Abstract.lines 15-30: Too much details in Abstract. Remove that WRF can be applief for both 3DVAR and 4DVAR and tell only what you used. Also, you do not need to tell all dates of experiments in Abstract.

Page 2. lines 23-24. Here I am confident that you have misinterpreted the findings of Lindskog et al. when it comes to benefit of 4DVAR aganst 3DVAR. As far as I can read, Lindskog et al. did not apply 4DVAR, only 3DVAR. They did an experiment with

modified background error statistics in 3DVAR and from the results they concluded that 'The assimilation of GNSS ZTD in NWP can benefit from more general data assimilation improvements, such as enhanced description of statistical information or improved data assimilation algorithms.'.

Page 2, line 19. ZTD stands for 'Zenith Total Delay' not 'Zenith Tropospheric Delay'?

Section 2.1, pages 4-6. Here you need to describe more details regarding the data assimilation setup. For example, is data assimilation only carried out in the inner domain or in both. How often are model runs started (once a day?) and when (00 UTC?). Is surface data assimilation applied and what kinds of background error representation and quality control is applied. In addition please justify the model set-up choices presented in Table 1, for example why you did not use cloud microphysics in Domain 2. Also refer to Papers presenting the details of the various schemes applied.

Section 3, methodology: The cost function can be derived from Bayesian probability theory as is done in: Lorenc, A., 1986, Analysis methods for numerical weather prediction. Q. J. R. Meteor. Soc., 112, 1177-1194. then you will see that a factor 1/2 is missing in equation 1 page 29.

In addition a reckomend to use standard notations defined in

Ide K, Courtier P, Ghil M, Lorenc A. 1997. Unified notation for data assimilation: Operational, sequen- tial and variational.J. Met. Soc. of Japan 75 : 181–189.

throughout the Paper. For example R instead of O.

page 8, line 23 Please do not use long subroutine names from model code, not so clear.

Section 2.3, Model evaluation: Here I see the main weakness of this Paper. The GNSS ZTD and PW in the first place mainly affect the 3-D distribution of humidity. That will affect rain later on. Modifying the initial humidity state will mainly influence short range forecasts due short predictability time scales and small model domain. In

addition statistical significance of results needs to be addresses.In fact already from Figures 4 in the Paper one can get a hint that one should not look for impact at ranges beyond 12-24 h. In my opinion verification scores should be re-derived using shorter forecast ranges and looking and the dependence on forecast range. In addition please look at what the data assimilation is doing at range 0 to start with and also look at forecast fields. For statistical verification do not look only at the surface but use the radiosondes you show you have in the domain for verification for different altitudes in the atmosphere. Please also prove confidence intervals to your results together with an explanation how these were derived.

---

## Referee Comment (RC2) · Anonymous Referee #2 · 13 Sep 2018

This manuscript presents Observing System Experiments (OSEs) with assimilation of GNSS Zenith Total Delay (ZTD) and Precipitable Water (PW) in the Weather Research and Forecasting model (WRF) over Poland for the period May-June 2013. The period was selected for a GNSS benchmark campaign during COST Action GNSS4SWEC and reported in the paper by Dousa et al (2016). To the best of my knowledge this is the only assimilation experiment conducted for this period and this makes the contribution of particular interest to the community. However, the benefits of the GNSS dataset collected during the benchmark campaign are not exploited fully (see 2) bellow), which is likely reflected in the results from OSEs.

[Figure]

Major comments/questions:

1) In the paper are used the "International GNSS Service (IGS) ultrarapid orbits, clocks and Earth rotation parameters are used." Please justify the selection of those products as their quality is likely having substantial impact on the OSEs. Please, provide a comparison to high quality near-real time estimates.

2) The quality control of GNSS ZTD is a vital part of the assimilation process. Please, include a section covering the quality control and "black listing" strategy you used. The benchmark quality controlled data-set can be used as a reference.

3) In section 4 is missing the model performance for PW (ZTD). It is not expected to improve the model if it has a very good PW (ZTD), which is likely the case for most of the time. Please, consider including a section with PW comparison of reference (REF) model (without assimilation) and GNSS PW.

4) The reported OSE impact do not cover assessment of PW improvement/degradation. It is important to access both individual positive and negative PW assimilation cases as they can provide valuable insight about the model and the ways to improve it.

5) The assimilation of GNSS data is limited to Poland while there are a large number of GNSS stations in the surrounding countries like Germany (over 500). For the large scale frontal processes the westerly flow modification (through data assimilation) is likely to be more valuable than the local modifications thus the question is if this has been considered. It is recommended to conducted OSEs for selected number of days with assimilation of GNSS data from the neighbouring countries.

Minor comments/questions:

1) Please consider revising the following paragraph in the abstract as it is not fully in line with the state of the art: "The GNSS data assimilation is currently widely discussed in the literature with respect to the various applications in meteorology and numerical
weather models. Data assimilation combines atmospheric measurements with knowledge of atmospheric behavior as codified in computer models. With this approach, the 'best' estimate of current conditions consistent with both information sources is produced. Some approaches allow assimilating also the non-prognostic variables, including remote sensing data from radar or GNSS (Global Navigation Satellite System). These techniques are named variational data assimilation schemes and are based on a minimization of the cost function, which contains the differences between the 15 model state (background) and the observations."

2) Page 1 line 27: Please specify if "20% improvement in bias of humidity forecast," is at surface or in 3D.

3) Page 2 line 21: Please correct "Authors".

4) Page 4 line 13: Please correct the colloquial language use in "PW into the very popular WRF model using the WRFDA package".

5) In section 4 it can be suggest to use the widely accepted terms REF run and OSE1, OSE2 etc instead of "base run".

6) It is not really clear what is displayed and how probability of detection and success ratio are computed in figure 3, 6, 7 and 8. The figures can be combined in one figure and referred as figure a), b), c) and d) as they show similar information. Figure captions are not of sufficient detail.

7) Page 16 line 10: Please explain why is ZTD impact much higher (43%) compared to PW (2%). "Relative humidity MEs are reduced by assimilation of PW by 2% and up to 43% while ZTD is used."

8) Page 18 line 9: "Adding SYNOP stations and radiosonde did not bring any further improvements in forecasting humidity or rain but reduced the errors in wind speed and temperature data." One reason can be that the driving initial and boundary conditions are with assimilated SYNOP and RS data. Please comment on this.

---

## Author Comment (AC1) · 24 Nov 2018

Anonymous Referee #1 General comments This Paper is concerned with evaluating the impact of GNSS observations within the WRF limited area data assimilation and modelling system over an area covering Poland. Previous work in the field of research is throughly reviewed. However, unfortunately, misinterpetations were found. The Paper has a sound scientific basis in the sense the effect of utilizing GNSS-based observations on the quality of numerical weather prediction is investigated. The experiments carried out are quite clearly described, although some improvements can be made. Unfortunately, in my opinion, the results of the experiments is not well enough evaluated and presented. Over such a small area and for a humidity related observations it is not advisable to evaluate the impact including forecasts up to 48 h and focus on surface observations and precipitation. In my opinion one should focus on forecasts up to roughly +12h and on verification also on upper air fields. In addition the statistical significance of the results should be presented. Major Revisions of the Paper are needed, in particular regarding the evaluation of the parallel experiment. After these have been carried out the Paper can be considered for scientific publication in Journal Atmos. Meas. Tech..

[WR] We are grateful for time and effort Reviewer has spent on the manuscript, we highly appreciate these comments. With analysis of majority of the weather events in May and June 2013 we actually see that these were regional – scale events: in May a westerly flows brought a series of precipitation and advection of cold air masses, in June, south east flow brought a humid and instable air masses. In both cases analysis of cases should be limited to 12-24h as the air volume that is observed by GNSS will be entirely replaced during one full day. Therefore analysis performed in this study should be either supported by observations from most of the countries to the West and South of Poland or limited in time to one day. We decide for the second as we have only full control over GNSS data from Poland, the processing strategies, whereas all data that are not process by WUEL E-GVAP might introduce biases that are not in agreement with the one introduced by WUEL. We modified therefore Tables 2 to 4 and Figures 4 and 5 (now Figure 7 and 8) to reflect this overall change.

Please find below more detailed comments and suggestions are listed. More Specific Comments Page 1. line 11: I suggest change 'codified' to 'represented'

[WR]Corrected

Abstract.lines 15-30: Too much details in Abstract. Remove that WRF can be applied for both 3DVAR and 4DVAR and tell only what you used. Also, you do not need to tell all dates of experiments in Abstract.

[WR] We decided to correct the details regarding available VAR schemes in WRF. However we would like to keep the exact dates of experiments as it is repeatedly used in the manuscript, more as an event label, and less as time indicators.

Page 2. lines 23-24. Here I am confident that you have misinterpreted the findings of Lindskog et al. when it comes to benefit of 4DVAR against 3DVAR. As far as I can read, Lindskog et al. did not apply 4DVAR, only 3DVAR. They did an experiment with modified background error statistics in 3DVAR and from the results they concluded that the assimilation of GNSS ZTD in NWP can benefit from more general data assimilation improvements, such as enhanced description of statistical information or improved data assimilation algorithms.'.

[WR] This is rather a problem with a following paragraph: "The results were mixed: for all cases the introduction of GPS ZTD increased the humidity bias, however the improvements of clouds forecasts were observed. Authors also identify no clear benefit of 4DVAR against 3DVAR. (Lindskog et al., 2017) in their Nordic country study of GNSS ZTD impact on forecasts, confirmed that the forecasts are sensitive to thinning distance." Which suggest that Lindskog et al. (2017) were testing 4DVAR against 3DVAR. However it should point to paper by Bennitt and Jupp, (2012) that were actually verifying impact of these two varational assimilation schemes. Therefore to clear this misunderstanding we change this paragraph to: "The results were mixed: for all cases the introduction of GPS ZTD increased the humidity bias, however the improvements of clouds forecasts were observed. (Bennitt and Jupp, 2012) also identify no clear benefit of 4DVAR against 3DVAR. (Lindskog et al., 2017) in their Nordic country study of GNSS ZTD impact on forecasts, confirmed that the forecasts are sensitive to thinning distance."

Page 2, line 19. ZTD stands for 'Zenith Total Delay' not 'Zenith Tropospheric Delay'? [WR] Well in literature both terms are used, however the first one is clearly related more to the meteorology (e.g. Bennitt and Jupp, 2012; Poli et al., 2007) while the second is more often is used in the navigation/positioning type of research (e.g. Lanyi, 1984;

[Figure]

Askne and Nordius, 1987; Ge et al., 2000). However the opposite is also possible (Vedel et al., 2004). We decide to change the "Zenith Tropospheric Delay" into "Zenith Total Delay"

Askne, J., & Nordius, H. (1987). Estimation of tropospheric delay for microwaves from surface weather data. Radio Science, 22(3), 379-386. Ge, M., Calais, E., & Haase, J. (2000). Reducing satellite orbit error effects in near real‐time GPS zenith tropospheric delay estimation for meteorology. Geophysical Research Letters, 27(13), 1915-1918. Bennitt, G. V., & Jupp, A. (2012). Operational assimilation of GPS zenith total delay observations into the Met Office numerical weather prediction models. Monthly Weather Review, 140(8), 2706-2719. Vedel, H., Huang, X. Y., Haase, J., Ge, M., & Calais, E. (2004). Impact of GPS zenith tropospheric delay data on precipitation forecasts in Mediterranean France and Spain. Geophysical research letters, 31(2). Lanyi, G. (1984). Tropospheric delay effects in radio interferometry. TDA Prog. Rep. 42-78, vol. April, 152-159. Poli, P., Moll, P., Rabier, F., Desroziers, G., Chapnik, B., Berre, L., ... & El Guelai, F. Z. (2007). Forecast impact studies of zenith total delay data from European near real‐time GPS stations in Météo France 4DVAR. Journal of Geophysical Research: Atmospheres, 112(D6).

Section 2.1, pages 4-6. Here you need to describe more details regarding the data assimilation setup. For example, is data assimilation only carried out in the inner domain or in both. How often are model runs started (once a day?) and when (00 UTC?). Is surface data assimilation applied and what kinds of background error representation and quality control is applied. In addition please justify the model set-up choices presented in Table 1, for example why you did not use cloud microphysics in Domain 2. Also refer to Papers presenting the details of the various schemes applied.

[JG] Data assimilation was run using 4DVAR WRF DA system, only for inner domain (d02). Prediction model was started once a day, at 00 UTC. Assimilation window was prepared at 00 UTC. Background Error covariance (BE) was selected for regional application (cv_options=5) ( BE depends on the WRF domain). BE was constructed based

on a forecast for convection event in the first week of May 2013. Quality control was selected for SYNOP and RS data in observation processor (obsproc) and in WRFDA. For ZTD and PWAT data, quality control was conducted during it prepare process, in obsproc and last step in 4DVAR assimilation. The WRF configuration based on the best ensemble members (ens1) with small modification from ensemble system dedicated for Poland area during summertime (Guzikowski, et al. 2015).

Guzikowski, J., Czerwińska, A. E., Krzyścin, J. W., & Czerwiński, M. A. (2017). Controlling sunbathing safety during the summer holidays-The solar UV campaign at Baltic Sea coast in 2015. Journal of Photochemistry and Photobiology B: Biology, 173, 271-281.

Section 3, methodology: The cost function can be derived from Bayesian probability theory as is done in: Lorenc, A., 1986, Analysis methods for numerical weather prediction. Q. J. R. Meteor. Soc., 112, 1177-1194. then you will see that a factor 1/2 is missing in equation 1 page 29.In addition a recommend to use standard notations defined in Ide K, Courtier P, Ghil M, Lorenc A. 1997. Unified notation for data assimilation: Operational, sequential and variational.J. Met. Soc. of Japan 75 : 181–189. throughout the Paper. For example R instead of O. [WR] Thank you for this valuable remark $\frac{1}{2}$ term was inserted to the equation 1, the notation now follows the recommended one by Ide et al., (1997).

page 8, line 23 Please do not use long subroutine names from model code, not so clear.

[WR] We would like to keep the module names, as for us finding the exact location of operator in the WRFDA code was very difficult and learning it from this manuscript could greatly improve further research on the code development (e.g. better parameterisation of ZHD).

Section 2.3, Model evaluation: Here I see the main weakness of this Paper. The GNSS ZTD and PW in the first place mainly affect the 3-D distribution of humidity.

That will affect rain later on. Modifying the initial humidity state will mainly influence short range forecasts due short predictability time scales and small model domain. In addition statistical significance of results needs to be addresses .In fact already from Figures 4 in the Paper one can get a hint that one should not look for impact at ranges beyond 12-24 h. In my opinion verification scores should be re-derived using shorter forecast ranges and looking and the dependence on forecast range. In addition please look at what the data assimilation is doing at range 0 to start with and also look at forecast fields. For statistical verification do not look only at the surface but use the radiosondes you show you have in the domain for verification for different altitudes in the atmosphere. Please also prove confidence intervals to your results together with an explanation how these were derived.

[MK] We agree with this comment and we have modified and extended the model evaluation by: Tables 2-4 have been changed. The error statistics were recalculated for the first 24h (it was 48h before). This is now clarified in text. The direct comparison between previous version of tables and current one can be find in the attachments. We have compared the forecasts with high-resolution radiosonde data. The radiosonde data were available for three stations in Poland (Wrocław, Warszawa and Łeba). 95% confidence intervals were added in the plots. Similar plots, but for relative humidity only, have been presented earlier by Guerova et al (2005). The confidence intervals were calculated by adding/substracting 1.96 * standard error to the mean error. We have provided the following plots for the 12h lead time forecast (left is for relative humidity, right for air temperature). However, adding the confidence intervals makes the plots hard to read, especially in the lower layers of the model. It clearly shows that the impact of ZTD is larger than IWV in both Temperature and Humidity. However for RH the impact is more significant especially between 2.5km and 10km.

Please also note the supplement to this comment:
https://www.atmos-meas-tech-discuss.net/amt-2018-263/amt-2018-263-AC1-supplement.pdf

---

## Author Comment (AC2) · 24 Nov 2018

This manuscript presents Observing System Experiments (OSEs) with assimilation of GNSS Zenith Total Delay (ZTD) and Precipitable Water (PW) in the Weather Research and Forecasting model (WRF) over Poland for the period May-June 2013. The period was selected for a GNSS benchmark campaign during COST Action GNSS4SWEC and reported in the paper by Dousa et al (2016). To the best of my knowledge this is the only assimilation experiment conducted for this period and this makes the contribution of particular interest to the community. However, the benefits of the GNSS dataset

collected during the benchmark campaign are not exploited fully (see 2) bellow), which is likely reflected in the results from OSEs.

[WR] Thank you for encouraging us to investigate COST Action GNSS4SWEC benchmark period with greater detail, hopefully by adding radiosonde observations and direct comparison of WRF derived PW and ZTD with GNSS-based counterparts we improved manuscript considerably.

Major comments/questions: 1) In the paper are used the "International GNSS Service (IGS) ultrarapid orbits, clocks and Earth rotation parameters are used." Please justify the selection of those products as their quality is likely having substantial impact on the OSEs. Please, provide a comparison to high quality near-real time estimates.

[WR] The data used in this study are covering May and June of 2013, and area of Poland it coincides with Benchmark campaign of COST Action GNSS4SWEC, but the data used in this study are directly taken from the NRT processing system as it was run in the 2013. The quality of retrieved troposphere parameters using the same setup as for data applied in this study are discussed in details in Bosy et al., (2012). We are not aware that there are other "higher quality near-real time estimates" as the 90 minutes latency constrain imposed on E-GVAP processing centre requires use of ultrarapid orbits, clocks and Earth rotation parameters. We are positive that application of the data as they were processed is a strong point of this paper, as it shows the exact impact one would get using operational (E-GVAP) product.

Bosy, J., Kaplon, J., Rohm, W., Sierny, J., & Hadas, T. (2012). Near real-time estimation of water vapour in the troposphere using ground GNSS and the meteorological data. In Annales Geophysicae (Vol. 30, No. 9, p. 1379). Copernicus GmbH.

2) The quality control of GNSS ZTD is a vital part of the assimilation process. Please, include a section covering the quality control and "black listing" strategy you used. The benchmark quality controlled data-set can be used as a reference.

[Figure]

[KW] The GNSS ZTDs are processed for the stations of European Position Determination System Active Geodetic Network (ASG-EUPOS, www.asgeupos.pl), which are continuously monitored for quality. Only the national reference stations of that system are taken into the assimilation. We added a reference in the data section. Moreover, we also adjust the formal errors for all stations by multiplying them by a factor of 10.5 mm (a standard deviation of the differences between WRF and ZTDs) and removing the observations (black listing) which errors exceed 20 mm.

3) In section 4 is missing the model performance for PW (ZTD). It is not expected to improve the model if it has a very good PW (ZTD), which is likely the case for most of the time. Please, consider including a section with PW comparison of reference (REF) model (without assimilation) and GNSS PW.

[WR] This is actually very good remark we extended the manuscript by adding additional section discussing comparison of GNSS based PW and WRF based PW

4) The reported OSE impact do not cover assessment of PW improvement/degradation. It is important to access both individual positive and negative PW assimilation cases as they can provide valuable insight about the model and the ways to improve it.

[KW] Thank you for the suggestion, it is also a very important aspect of GNSS data assimilation. We added the comparisons with the GNSS products, PW and ZTD with WRF before and after the assimilation.

5) The assimilation of GNSS data is limited to Poland while there are a large number of GNSS stations in the surrounding countries like Germany (over 500). For the large scale frontal processes the westerly flow modification (through data assimilation) is likely to be more valuable than the local modifications thus the question is if this has been considered. It is recommended to conducted OSEs for selected number of days with assimilation of GNSS data from the neighbouring countries.

[Figure]

[MK] We have re-investigated the literature and OSEs for assimilation of GNSS, it is clear that Lindskog et al., (2017) were taking observations across whole Scandinavia; Bennitt and Jupp (2012) were using observations from UK and Netherlands with large density and sparse across Europe, and Poli et al. (2007) adopted in their study pan European network. This allowed to modify large scale flows e.g. westerly flow as it passes over Europe for few days up to 3 days. However, in case presented in this manuscript what we have is only small section of European troposphere covered by GNSS observations mainly Poland. Therefore we agree with the reviewer to study impact of GNSS data on longer forecasts than 12-24h, one would need to extend the area covered with observations. In our case this is not easy achievable, as we don't have available GNSS data from other countries of Europe beside Poland in 2013. Even attaining access to the troposphere retrieval for the selected time period and location (May and June, 2013 in Europe), will result in non-uniform processing strategies and hence quality of assimilated GNSS data in two ends of the WRF model. Therefore we decided to reduce the studied impact to 24hours which is reasonable assumption, keeping in mind the typical weather situation in May and June as well horizontal extend of Poland.

Minor comments/questions:

1) Please consider revising the following paragraph in the abstract as it is not fully in line with the state of the art: "The GNSS data assimilation is currently widely discussed in the literature with respect to the various applications in meteorology and numerical weather models. Data assimilation combines atmospheric measurements with knowledge of atmospheric behavior as codified in computer models. With this approach, the 'best' estimate of current conditions consistent with both information sources is produced. Some approaches allow assimilating also the non-prognostic variables, including remote sensing data from radar or GNSS (Global Navigation Satellite System).These techniques are named variational data assimilation schemes and are based on a minimization of the cost function, which contains the differences between the model state (background) and the observations."

[WR] The following sentence was added: "The variational assimilation is a first choice for data assimilation in the weather forecast centres, however current research is consequently looking into use of iteratice, filtering approach such as Extended Kalman Filter(EKF).

2) Page 1 line 27: Please specify if "20% improvement in bias of humidity forecast," is at surface or in 3D.

[WR] This improvement is visible in the surface data as in the initial submission we only considered for validation SYNOP observations. In the revised manuscript we also used for validation two other data sources GNSS and Radiosondes.

3) Page 2 line 21: Please correct "Authors".

[WR] Corrected to: "(Bennitt and Jupp, 2012)"

4) Page 4 line 13: Please correct the colloquial language use in "PW into the very popular WRF model using the WRFDA package".

[WR] Corrected to: "assimilation of GPS ZTD and PW into widely adopted WRF model using the WRFDA package."

5) In section 4 it can be suggest to use the widely accepted terms REF run and OSE1, OSE2 etc instead of "base run".

[WR] We understand the need to keep the manuscript language to mostly used in the data assimilation community terms, however this study is on the edge between GNSS and assimilation community and all labels as wells as reference in the manuscript were called "base run".

6) It is not really clear what is displayed and how probability of detection and success ratio are computed in figure 3, 6, 7 and 8. The figures can be combined in one figure and referred as figure a), b), c) and d) as they show similar information. Figure captions

are not of sufficient detail.

[MK] We would like to refrain from merging these figures, it is already quite difficult to keep the number of observations visible on all figures (with often overlap between rain rates with and without assimilation). The performance diagrams are tool that help to identify whether the improvement after assimilation of observations. The easiest first glance approach is to verify whether the marked data sets are close to the intersection line and close to the top right corner.

7) Page 16 line 10: Please explain why is ZTD impact much higher (43%) compared to PW (2%). "Relative humidity MEs are reduced by assimilation of PW by 2% and up to 43% while ZTD is used."

[WR]This might be an side-effect to alternation by the ZTD operator not only the humidity like PW, but also pressure and temperature. Which in turn alternates advection of the humidity.

8) Page 18 line 9: "Adding SYNOP stations and radiosonde did not bring any further improvements in forecasting humidity or rain but reduced the errors in wind speed and temperature data." One reason can be that the driving initial and boundary conditions are with assimilated SYNOP and RS data. Please comment on this.

[JG] You remark is correct, we used in this study initial and boundary conditions from National Center for Environmental Prediction Final Analysis, Operational Model Global Tropospheric Analyses and it is widely known that these data contain radiosonde observations and SYNOPs, however the exact number and names of observations taken into account are not available for public discussion.